

# Extended Bloch-McConnell equations for mechanistic analysis of hyperpolarized $^{13}$C magnetic resonance experiments on enzyme systems

Thomas R. Eykyn[1], Stuart J. Elliott[2,4] and Philip W. Kuchel[3]

[1] School of Biomedical Engineering and Imaging Sciences, King's College London, St Thomas' Hospital, London SE1 7EH, United Kingdom
[2] Centre de Résonance Magnétique Nucléaire à Très Hauts Champs - FRE 2034 Université de Lyon / CNRS / Université Claude Bernard Lyon 1 / ENS de Lyon, 5 Rue de la Doua, 69100 Villeurbanne, France
[3] School of Life and Environmental Sciences, University of Sydney, NSW 2006, Australia
[4] Current address: Department of Chemistry, University of Liverpool, Liverpool L69 7ZD, United Kingdom

*Correspondence to*: Thomas R. Eykyn (thomas.eykyn@kcl.ac.uk)

This article is dedicated to Geoffrey Bodenhausen on the occasion of his 70[th] Birthday.

**Abstract.** We describe an approach to formulating the kinetic master equations of the time evolution of NMR signals in reacting (bio)chemical systems. Special focus is given to studies that employ signal enhancement (hyperpolarization) methods such as dissolution dynamic nuclear polarization (dDNP) and involving nuclear spin-bearing solutes that undergo reactions mediated by enzymes and membrane transport proteins. We extend the work given in a recent presentation on this topic to now include enzymes with two or more substrates and various enzyme reaction mechanisms as classified by Cleland. Using this approach, we can address some pressing questions in the field from a theoretical standpoint. For example, why does binding of a hyperpolarized substrate to an enzyme *not* cause an appreciable loss of the signal from the substrate or product? Why does the concentration of an unlabelled pool of substrate, for example $^{12}$C lactate, cause an increase in the rate of exchange of the $^{13}$C labelled pool? To what extent is the equilibrium position of the reaction perturbed during administration of the substrate? The formalism gives a full mechanistic understanding of the time courses derived and is of relevance to ongoing clinical trials using these techniques.

## 1 Introduction

Nuclear magnetic resonance (NMR) spectroscopy and imaging (MRI) are widely employed techniques with far-reaching applications in physics, chemistry, medicine and the life sciences. NMR and MRI provide a wealth of information from structure elucidation, protein dynamics and metabolic profiling through to disease diagnostics in oncology, cardiology and neurology among others. The technique's low sensitivity is one of the primary concerns in the magnetic resonance community and is often a limiting factor in experiments from solid-state NMR to medical imaging. Recent work has shown that the sensitivity of NMR experiments can be improved by using non-equilibrium hyperpolarization techniques such as dissolution dynamic nuclear polarization (dDNP) to boost signal intensities by many orders of magnitude (Ardenkjaer-Larsen et al., 2003). Such techniques have led to new applications (Golman et al., 2003; Golman et al., 2006; Keshari and Wilson, 2014) and necessitated the development of acquisition strategies to exploit the hyperpolarized magnetization in a time efficient manner (Yen et al., 2009); as well as new tools for signal processing and image reconstruction (Hu et al., 2010). A challenge





with the interpretation of these recordings is that, unlike radiotracers, hyperpolarized MR is a non-tracer technique
requiring the injection of physiological or even supra-physiological concentrations of substrate.

To date there have been many mathematical methods devised for analyzing the kinetic time courses in

dDNP NMR studies (Zierhut et al., 2010; Hill et al., 2013b; Pagès and Kuchel, 2015; Daniels et al., 2016).
However, until recently there has been little consensus on the best methods for analyzing and then interpreting
reaction kinetics measured therein. A theoretical framework has only recently appeared to fully elucidate the
underlying mechanisms (Kuchel and Shishmarev, 2020). One challenge is that the widely used Bloch-McConnell
equations describe the exchange of magnetization of only the MR active nuclei while the reaction kinetics are
subject to a plethora of molecular interactions in a (bio)chemical milieu. Furthermore, in a typical hyperpolarized
MR experiment the initial injection of a non-tracer concentration of substrate causes the reaction system to be
perturbed from its equilibrium state, or quasi-steady state, and therefore the concentrations of the reactants are
time dependent. In this regard, challenges relate to the description of non-linear kinetics, for example second order
reactions, and the involvement of un-observable (non-labelled) metabolites to the overall kinetics, *e.g.,* enzyme
cofactors, co-substrates and natural abundance $^{12}$C-containing metabolites (Hill et al., 2013a); as well as explicit
descriptions of enzyme mechanisms *e.g.,* sequential ordered, sequential random, double displacement (ping-pong)
reactions, and allosteric interactions that occur on an enzyme far from its active site. Enzyme activity is also
influenced by inhibitors that can be competitive, non-competitive, or uncompetitive (Cook and Cleland, 2007;
Cleland, 1967). Mathematical models of enzyme systems should agree with standard descriptions of (bio)chemical
kinetics while remaining capable of describing the time evolution of magnetization that is described by the Bloch-
McConnell equations (McConnell, 1958).

Here we address these issues in a stepwise manner, by developing a mechanistic approach that combines

the MR interactions with the chemical and/or enzyme mediated reactions described by the Bloch-McConnell
equations. These equations are grounded in the concept of conservation of mass of the hyperpolarized signal plus
its non-hyperpolarized counterpart and the various products; this was recently highlighted (Kuchel and
Shishmarev, 2020) where the MR visible signal decays to produce an MR invisible one such that the sum is
constant and proportional to the total solute concentration.

### 1.1 Basic concepts – sensitivity

We begin addressing the problem by defining the signal-to-noise ratio (SNR) in MR. In its most basic form,
sensitivity is described by the ratio of the signal amplitude divided by the root mean square of the amplitude of
the noise. When a signal $S(t)$ is detected in the NMR receiver coil that surrounds the sample, the magnitude of
the induced current is a function of: (*i*) the perturbation of nuclear spin populations from thermal equilibrium
$S_{sample}(t)$; plus (*ii*) a random contribution from the noise in the electronic circuitry $S_{electronics}(t)$. Hence:

$$S(t) = S_{sample}(t) + S_{electronics}(t) \quad . \tag{1}$$


The current induced in the coil is time-dependent and proportional to the magnetization that precesses in the *x,y*-
plane. In other words, the signal $S(t)$ is recorded until decoherence renders $S_{sample}(t)$ undetectable against the



noise, $S_{electronics}(t)$. The latter is mainly attributed to the radiofrequency (RF) circuitry in the probe head and the
preamplifier(s) (*e.g.,* Johnson noise (Johnson, 1928)) of the spectrometer. If the NMR signal (free induction decay;
FID) that is detected in a subsequent experiment is indistinguishable from the first, and the two are added together,
then the signal amplitude (peak area) will scale linearly with the number of added FIDs, $N$. The noise associated
with each experiment is random, and assuming its source remains fixed over time, *i.e.,* stationary noise, then the
amplitude scales with the square root of the number of FIDs, $N^{1/2}$. Hence signal summation enhances the SNR
of an NMR experiment in proportion to the square root of the number of FIDs. In other words, to achieve an
enhancement by a factor $\xi$ requires an increase in experiment duration of $\xi^2$. Therefore, unavoidably, FID
summation is a slow process and experiments can sometimes take days or weeks to achieve a sufficient SNR from
a sample of a low sensitivity nuclide or one with a long relaxation time. The amount of attainable signal averaging
is constrained when monitoring dynamic processes by NMR spectroscopy; and an inherently good SNR is
required from the outset for a time course experiment.

**1.2 Thermal effects**

89   The usual way to proceed when calculating the NMR response of a spin system to RF pulse sequences

is to solve the ordinary quantum mechanical master equation that describes the evolution of the spin density
operator (Hore et al., 2015). This is the Liouville-von Neumann equation, that has been extended to include non-
coherent interactions (predominantly relaxation phenomena) (Ernst et al., 1987):

$$\frac{d}{dt}\rho = -i\hat{H}\rho - \hat{\Gamma}(\rho - \rho_0) \quad , \tag{2}$$


where $\hat{H}$ is the commutation superoperator of the coherent Hamiltonian $H$ given by $\hat{H}\rho = [H,\rho]$, which contains
information on all spin-spin and field-spin interactions; while $\hat{\Gamma}$ is the relaxation superoperator that describes all
longitudinal ($T_1$) and transverse ($T_2$) relaxation processes, as well as any cross-relaxation or cross-correlation
interactions. Note, that in the interests of reducing clutter in equations (for which the operator context should be
clear) hereafter we have omitted carets denoting operators and only used them to denote superoperators.

100   Our aim here is to describe the kinetics of exchange between different solutes that contain hyperpolarized

nuclei *e.g.*, A $\leftrightarrow$ B, in which the relaxation times are constant. In this quest, the first simplifying assumption that
is worth exploring is that all intermolecular interactions, notably, scalar coupling, dipolar coupling, cross-
relaxation and cross-correlation between species A and B can be ignored. This applies to non-interacting solute
molecules in solution in which motional averaging occurs; and we focus on thermal effects on the evolution of
the FID.

106   The so-called Zeeman polarization term describes the sensitivity of $S_{sample}(t)$ in Eq. (1) to temperature

and magnetic field in an NMR experiment. Magnetic polarization is described by the equilibrium density operator
$\rho_0$ that specifies the probability distribution of states. Zeeman polarization corresponds to the magnitude of
normalized longitudinal spin order $I_z$ that is contained in $\rho_0$. Specifically, for an ensemble of spin-½ nuclei this
is given by (Ernst et al., 1987):



$$\rho_0 = \frac{exp(-\hbar H_0/kT)}{Tr\{exp(-\hbar H_0/kT)\}} \quad , \tag{3}$$

where $k$ is the Boltzmann constant and $T$ is the temperature (Kelvin). The Zeeman Hamiltonian $H_0$ describes the interaction of the spins with the static magnetic field of magnitude $B_0$, given by $H_0 = \omega_0 I_z$, where $\omega_0$ is the Larmor frequency (rad s$^{-1}$). In the basis of the two eigenstates $|\alpha\rangle$ ("spin-up") and $|\beta\rangle$ ("spin-down"), the equilibrium density operator is written in matrix form as:

$$\rho_0 = \frac{1}{Z}\begin{bmatrix} exp(\hbar\omega_0/2kT) & 0 \\ 0 & exp(-\hbar\omega_0/2kT) \end{bmatrix} \quad , \tag{4}$$

where $Z$ is the partition function, given by $Z = \sum_{i=1}^{M} exp(-\varepsilon_i/kT)$, and $M$ is the number of states ($M = 2$ for an $I = \frac{1}{2}$ nucleus). In the case of a spin-½ system, the partition function is the sum of the populations $Z = exp(\hbar\omega_0/2kT) + exp(-\hbar\omega_0/2kT) \approx 2$ when $\varepsilon_i$ is very small, as is typically the case at thermal equilibrium in NMR systems. The Zeeman polarization is proportional to the projection of the spin density operator onto the angular momentum operator. In other words, it is proportional to the expectation value of $\langle I_z \rangle$, and is given by (Keeler, 2010):

$$\langle I_z \rangle = Tr[\rho_0 \, I_z] = \frac{1}{2Z}[exp(\hbar\omega_0/2kT) - exp(-\hbar\omega_0/2kT)] \quad . \tag{5}$$

Hence, the Zeeman polarization for an ensemble of nuclear spins is the normalized *imbalance* between the populations of the $|\alpha\rangle$ and $|\beta\rangle$ states, $p_\alpha$ and $p_\beta$, respectively; in other words, it is the normalized net population difference that is given by:

$$P = \frac{p_\alpha - p_\beta}{p_\alpha + p_\beta} \quad . \tag{6}$$

This normalization is carried out with respect to the total population of the nuclear ensemble such that $p_\alpha + p_\beta = 1$. Therefore, the bounds on the polarization are $-1 < P < +1$. At room temperature (~298 K), and in a field of 11.75 T (500 MHz for $^1$H nuclei), the thermal equilibrium Zeeman polarization, $P_{z,eq}$, is a mere ~$4 \times 10^{-5}$. Thus, there is only a tiny population difference between the spin states of a nuclear ensemble that implies inherently weak polarization. It is this small population imbalance which is manipulated in NMR experiments under thermal equilibrium conditions. This weak polarization is a consequence of the small difference in energy (~0.1 J mol$^{-1}$) between nuclear spin energy levels at room temperature (~2.5 kJ mol$^{-1}$); and it implies only weak alignment of nuclear spins in the static magnetic field of all contemporary superconducting magnets.

In the usual quantum mechanical analysis of multiple spin systems, the density operator (that describes the probability density of states) is normalized to 1, meaning that the summed (total) probability density of all states is 1. This is expressed mathematically as $Tr[\rho] = 1$, where $Tr$ denotes the trace of the matrix (Hore et al., 2015). To describe non-equilibrium reactions in terms of solute concentrations requires a scaled density operator (Kuhne et al., 1979):

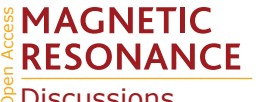


$$\sigma_i = [A_i]\rho_i \quad , \tag{7}$$


where $\sigma_i$ is now proportional to [A$_i$]. Differentiation of Eq. (7) leads to:

$$\frac{d\sigma_i}{dt} = [A_i]\frac{d\rho_i}{dt} + \frac{d[A_i]}{dt}\rho_i \quad . \tag{8}$$


Therefore, it follows that for a system at chemical equilibrium $d[A_i]/dt = 0$, so the scaled density operator is
directly proportional to the normalised density operator. For non-equilibrium systems the concentrations are time
dependent *viz.*, $d[A_i]/dt \neq 0$ so the two no longer scale in a straightforward manner.

153    On the other hand, equilibrium magnetization ($M_{z,eq}$) is a bulk property that is the net magnetic dipole

moment per unit volume; and is proportional to $\langle I_z \rangle$ where the proportionality factor is $N\hbar\gamma$. From Eq. (5) this
yields the expression for the magnetization in terms of magnetic field strength, temperature and number of spins
in the sample (or more specifically in the detection volume of the NMR spectrometer):

$$M_{z,eq} = \frac{N\hbar\gamma}{2} tanh\left(\frac{\hbar\gamma B_0}{2kT}\right) \quad . \tag{9}$$


In the so-called 'high temperature limit' (room temperature, in the cases addressed here) Eq. (9) simplifies to:

$$M_{z,eq} = \frac{N\hbar^2\gamma^2 B_0}{4kT} \quad . \tag{10}$$


In words, 'thermal magnetization' is proportional to the magnitude of the external magnetic field strength, $B_0$,
and is inversely proportional to the temperature, $T$, while being proportional to the number of spins, $N$. Therefore,
it is *proportional* to the concentration [A$_i$] of the solute that bears the NMR-active nucleus.

**2 Equation of motion – the Bloch equations**

167    In the absence of intermolecular binding (however transient), or scalar couplings, the motion (time

evolution) of magnetizations is described by the Bloch equations. Magnetization is explicitly declared to be
proportional to reactant concentrations [A] and [B], as has recently been discussed (Kuchel and Shishmarev,
2020). To explore this situation, we start with the basic Bloch equations for a single spin-½ ensemble. The
equation describes the time evolution of *x, y* and *z* magnetization in the rotating frame, and includes the influence
of chemical shift, RF fields, and transverse ($T_2$) and longitudinal relaxation ($T_1$) time constants. The Bloch
equations in their complete form are described as being inhomogeneous, and they can be written using a matrix
and vectors:





$$\frac{d}{dt}\begin{bmatrix} M_x \\ M_y \\ M_z \end{bmatrix} = -\begin{bmatrix} R_2 & \Omega & -\omega_y \\ -\Omega & R_2 & \omega_x \\ \omega_y & -\omega_x & R_1 \end{bmatrix}\begin{bmatrix} M_x \\ M_y \\ M_z \end{bmatrix} + \begin{bmatrix} 0 \\ 0 \\ R_1 M_{z,eq} \end{bmatrix} \quad , \tag{11}$$


where $\Omega = \omega_0 - \omega_{RF}$ is the 'offset frequency' in the rotating frame; $\omega_0$ (rad s⁻¹) is the Larmor frequency; $\omega_{RF}$
(rad s⁻¹) is the RF frequency; the *x* component of the RF field (rad s⁻¹) is $\omega_x = -\gamma B_1 \cos(\omega_{RF} t + \varphi)$; and the *y*
component is $\omega_y = -\gamma B_1 \sin(\omega_{RF} t + \varphi)$, where the magnitude of the field strength is $B_1$, and the phase of the
wave form relative to an internal reference source is $\varphi$. The longitudinal relaxation rate constant is denoted by
$R_1 = 1/T_1$; the transverse one by $R_2 = 1/T_2$; and the equilibrium magnetization by $M_{z,eq}$.

Equation (11) is tedious to solve analytically, but it is readily solved numerically (Allard et al., 1998;

Helgstrand et al., 2000). On the other hand, by including the identity operator in the basis set and adding a constant
to the equilibrium magnetization (Levitt and Dibari, 1992), we obtain a much more compliant (to analysis) matrix
equation:

$$\frac{d}{dt}\begin{bmatrix} \frac{E}{2} \\ M_x \\ M_y \\ M_z \end{bmatrix} = -\begin{bmatrix} 0 & 0 & 0 & 0 \\ 0 & R_2 & \Omega & -\omega_y \\ 0 & -\Omega & R_2 & \omega_x \\ -2\Theta & \omega_y & -\omega_x & R_1 \end{bmatrix}\begin{bmatrix} \frac{E}{2} \\ M_x \\ M_y \\ M_z \end{bmatrix} \quad , \tag{12}$$


where $E$ is equal to 1 and the factor $\Theta = R_1 M_{z,eq}$ describes the equilibrium magnetization.

**2.1 Chemical exchange kinetics of systems prior to and at equilibrium – the Bloch-McConnell equations**

We can extend the system of equations from describing an ensemble of single spins to two or more

exchanging spins. The system of equations now accounts for the magnetization interaction with the lattice and
exchange via the forward and reverse chemical reactions. These are the Bloch-McConnell equations (McConnell,

1958).

First, consider the rate expressions for a simple bi-directional chemical reaction. The coupled differential

equations describing first-order reaction kinetics of solute A becoming solute B and back again, A $\leftrightarrow$ B, are
typically expressed in terms of molar concentrations:

$$\frac{d[A(t)]}{dt} = -k_1[A(t)] + k_{-1}[B(t)] \quad , \tag{13}$$

$$\frac{d[B(t)]}{dt} = k_1[A(t)] - k_{-1}[B(t)] \quad , \tag{14}$$


that can be expressed in matrix form:

$$\frac{d}{dt}\begin{bmatrix} [A(t)] \\ [B(t)] \end{bmatrix} = \begin{bmatrix} -k_1 & k_{-1} \\ k_1 & -k_{-1} \end{bmatrix}\begin{bmatrix} [A(t)] \\ [B(t)] \end{bmatrix} \quad . \tag{15}$$






The rate constant for the forward reaction is denoted by $k_1$ while for the reverse reaction it is $k_{-1}$. The time
dependent concentrations are given by $[A(t)]$ and $[B(t)]$. As required by the fact that this is a closed system, the
equations must conform to the *principle of conservation of mass*. Specifically, the sum of the rates of change of
$[A(t)]$ and $[B(t)]$ given by $d[A(t)]/dt + d[B(t)]/dt$, is zero. We return to this point below. In other words,
mass is neither created nor destroyed during the reaction in such a closed system.
For the simplest case of two magnetically active solutes, each possessing a single spin-½ nuclide, in
chemical exchange, $A \leftrightarrow B$, the direct product (a mathematical operation used in quantum mechanics to generate
the necessary combinations of states) of the chemical (solute) space $\{[A], [B]\}$ and the magnetization vector space
$\{M_x, M_y, M_z\}$ for each of A and B is given by:

$$
\begin{bmatrix} 1 \\ 1 \end{bmatrix} \otimes \begin{bmatrix} M_x \\ M_y \\ M_z \end{bmatrix} = \begin{bmatrix} M_x^A \\ M_y^A \\ M_z^A \\ M_x^B \\ M_x^B \\ M_x^B \end{bmatrix} . \tag{16}
$$


A new exchange matrix in the basis of the new magnetization space $\{M_x^A, M_y^A, M_z^A, M_x^B, M_y^B, M_z^B\}$ is calculated by
taking the direct product of the exchange matrix with the identity operator $I$ that is chosen to have the same
dimensions as the magnetization space. The direct product is given by:

$$
\begin{bmatrix} -k_1 & k_{-1} \\ k_1 & -k_{-1} \end{bmatrix} \otimes \begin{bmatrix} 1 & 0 & 0 \\ 0 & 1 & 0 \\ 0 & 0 & 1 \end{bmatrix} = \begin{bmatrix} -k_1 & 0 & 0 & k_{-1} & 0 & 0 \\ 0 & -k_1 & 0 & 0 & k_{-1} & 0 \\ 0 & 0 & -k_1 & 0 & 0 & k_{-1} \\ k_1 & 0 & 0 & -k_{-1} & 0 & 0 \\ 0 & k_1 & 0 & 0 & -k_{-1} & 0 \\ 0 & 0 & k_1 & 0 & 0 & -k_{-1} \end{bmatrix} . \tag{17}
$$


Likewise, the matrix describing coherent and incoherent magnetization interactions can be recast in a similar
fashion to give:

$$
\begin{bmatrix} 1 & 0 \\ 0 & 1 \end{bmatrix} \otimes \begin{bmatrix} R_2 & \Omega & -\omega_y \\ -\Omega & R_2 & \omega_x \\ \omega_y & -\omega_x & R_1 \end{bmatrix} = \begin{bmatrix} R_2^A & \Omega^A & -\omega_y & 0 & 0 & 0 \\ -\Omega^A & R_2^A & \omega_x & 0 & 0 & 0 \\ \omega_y & -\omega_x & R_1^A & 0 & 0 & 0 \\ 0 & 0 & 0 & R_2^B & \Omega^B & -\omega_y \\ 0 & 0 & 0 & -\Omega^B & R_2^B & \omega_x \\ 0 & 0 & 0 & \omega_y & -\omega_x & R_1^B \end{bmatrix} . \tag{18}
$$


The inhomogeneous form of the Bloch equations can now be constructed to take into account both the coherent
and incoherent interactions, *as well as* chemical exchange. This yields the inhomogeneous form of the Bloch-
McConnell equations, which are written (again in matrix form) as:





$$\frac{d}{dt}\begin{bmatrix} M_x^A \\ M_y^A \\ M_z^A \\ M_x^B \\ M_x^B \\ M_x^B \end{bmatrix} = \begin{bmatrix} R_2^A + k_1 & \Omega^A & -\omega_y & -k_{-1} & 0 & 0 \\ -\Omega^A & R_2^A + k_1 & \omega_x & 0 & -k_{-1} & 0 \\ \omega_y & -\omega_x & R_1^A + k_1 & 0 & 0 & -k_{-1} \\ -k_1 & 0 & 0 & R_2^B + k_{-1} & \Omega^B & -\omega_y \\ 0 & -k_1 & 0 & -\Omega^B & R_2^B + k_{-1} & \omega_x \\ 0 & 0 & -k_1 & \omega_y & -\omega_x & R_1^B + k_{-1} \end{bmatrix} \begin{bmatrix} M_x^A \\ M_y^A \\ M_z^A \\ M_x^B \\ M_y^B \\ M_z^B \end{bmatrix} + \begin{bmatrix} 0 \\ 0 \\ R_1^A M_{z,eq}^A \\ 0 \\ 0 \\ R_1^B M_{z,eq}^B \end{bmatrix} , \qquad (19)$$

where $M_{z,eq}^A$ and $M_{z,eq}^B$ denote the respective equilibrium magnetizations (hence the subscript $eq$).

The inhomogeneous form of the Bloch-McConnell equations can similarly be modified by incorporating the equilibrium magnetization to create a homogeneous form of this master equation:

$$\frac{d}{dt}\begin{bmatrix} \frac{E}{2} \\ M_x^A \\ M_y^A \\ M_z^A \\ M_x^B \\ M_y^B \\ M_z^B \end{bmatrix} = \begin{bmatrix} 0 & 0 & 0 & 0 & 0 & 0 & 0 \\ 0 & R_2^A + k_1 & \Omega^A & -\omega_y & -k_{-1} & 0 & 0 \\ 0 & -\Omega^A & R_2^A + k_1 & \omega_x & 0 & -k_{-1} & 0 \\ -2\Theta^A & \omega_y & -\omega_x & R_1^A + k_1 & 0 & 0 & -k_{-1} \\ 0 & -k_1 & 0 & 0 & R_2^B + k_{-1} & \Omega^B & -\omega_y \\ 0 & 0 & -k_1 & 0 & -\Omega^B & R_2^B + k_{-1} & \omega_x \\ -2\Theta^B & 0 & 0 & -k_1 & \omega_y & -\omega_x & R_1^B + k_{-1} \end{bmatrix} \begin{bmatrix} \frac{E}{2} \\ M_x^A \\ M_y^A \\ M_z^A \\ M_x^B \\ M_y^B \\ M_z^B \end{bmatrix} . \qquad (20)$$

Again, the factors $\Theta^A = R_1^A M_{z,eq}^A$ and $\Theta^B = R_1^B M_{z,eq}^B$ account for the respective equilibrium magnetizations.

### 2.1.1 Simulations of thermal kinetics using Eq. (19)

Next, consider Eq. (19) for simulating the evolution of the $x$, $y$, and $z$ components of the magnetization of a 'thermal magnetization' (*non-hyperpolarized*) sample. We seek the NMR spectrum that results from a two-site exchange reaction between solutes A and B, Fig. 1(a), as conventionally observed in room temperature NMR experiments.

Simulations were performed in *MatLab* with an initial equilibrium magnetization of $\mathbf{M}_0 = \mathbf{M}_{eq} = [0.0, 1.0, 0, 0, 0.8]$ where $M_{z,eq}^A = 1.0$ and $M_{z,eq}^B = 0.8$ are the respective equilibrium $z$ magnetizations. Chemical shifts offsets were $\Omega^A = 10 \times 2\pi$ rad s$^{-1}$ and $\Omega^B = 10 \times 2\pi$ rad s$^{-1}$. Relaxation rate constants were $R_1^A = R_1^B = 1\ s^{-1}$ and $R_2^A = R_2^B = 1\ s^{-1}$. The influence of an RF$_y$ pulse was then calculated with $\omega_x = -\gamma B_1 \cos(\pi/2)$ and $\omega_y = -\gamma B_1 \sin(\pi/2)$ and with a field strength of 1.5 kHz, corresponding to $\omega_y = -\gamma B_1 = -1500 \times 2\pi$ rad s$^{-1}$ and $\omega_x = 0$. For a 90° RF nutation (flip) angle the pulse duration is $t_p = \pi/2\omega_y$, which gave a transformed magnetization vector after the pulse of $M(t) = [0.999, 0.007, 0.000, 0.800, -0.005, 0.000]$; this was composed mostly of $M_x^A + M_x^B$ with a residual contribution from $M_y^A + M_y^B$ arising from evolution of the chemical shift during the RF pulse; and a small contribution from $M_z^A + M_z^B$ due to return of the magnetization to the equilibrium state.

The observable signal (the FID, which is a function of time) is proportional to the complex signal $S(t) = M_x^A(t) - iM_y^A(t) + M_x^B(t) - iM_y^B(t)$. Noise was simulated by adding to the FID a normally distributed complex random vector with mean = 0 and standard deviation (SD) = 0.1. The spectrum $s(\omega)$ was then calculated by taking

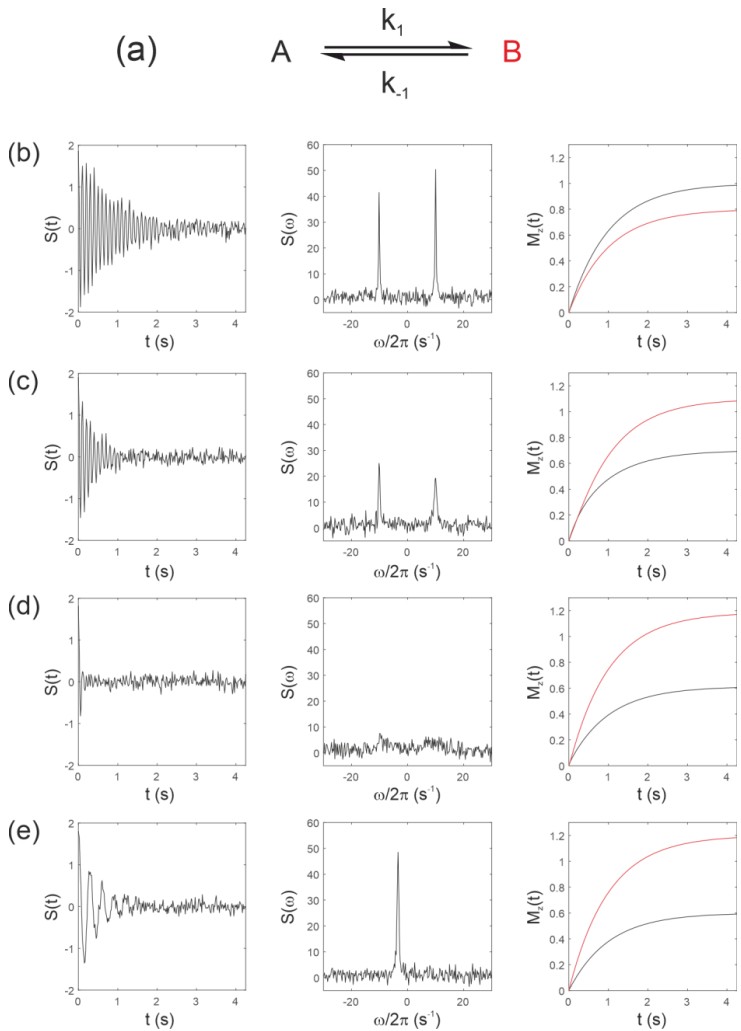

**Figure 1** Simulated NMR spectra resulting from a two-site exchange process between *thermally polarized* solutes, A ↔ B, shown schematically in (a). Simulated FIDs $S(t)$ are shown in (b-e) left panel, with corresponding spectra $s(\omega)$, middle panel, and the recovery of $z$ magnetizations, $M_z^A(t)$ and $M_z^B(t)$, right panel. Spectra were simulated with rate constants, (b) $k_1 = k_{-1} = 0$; (c) $k_1 = 2\ s^{-1}$, $k_{-1} = 1\ s^{-1}$; (d) $k_1 = 20\ s^{-1}$, $k_{-1} = 10\ s^{-1}$; and (e) $k_1 = 2000\ s^{-1}$, $k_{-1} = 1000\ s^{-1}$, corresponding to no exchange, slow, intermediate, and fast exchange regimes, respectively.

the Fourier transform of $S(t)$. Simulated FIDs $S(t)$ are shown in Figs. 1(b-e) left panel, the corresponding spectra
$s(\omega)$ in Figs. 1(b-e) middle panel, and the recovery of the $z$ magnetizations $M_z^A(t)$ and $M_z^B(t)$ are shown in Figs.
1(b-e), right panel. Spectra were simulated for a range of rate constants, where exchange was either absent $k_1 =$
$k_{-1} = 0$, Fig. 1(b); or for increasing rates of exchange. Thus, (c) $k_1 = 2\ s^{-1}$, $k_{-1} = 1\ s^{-1}$; (d) $k_1 = 20\ s^{-1}$,
$k_{-1} = 10\ s^{-1}$; and (e) $k_1 = 2000\ s^{-1}$, $k_{-1} = 1000\ s^{-1}$, corresponding to the slow, intermediate and fast
regimes, respectively.




The equilibrium constant was fixed so that $K = k_1/k_{-1} = 2$; hence the system was not at chemical

equilibrium at $t = 0\ s$. The simulations highlight an important point: In the absence of exchange the Bloch-

McConnell equations predict the recovery of the $z$ magnetizations back to their equilibrium values $M_{z,eq}^A$ and $M_{z,eq}^B$

while under conditions of fast exchange this no longer holds, and a non-equilibrium system will rapidly return to

its chemical equilibrium, not to its thermal equilibrium, within the timescale of the NMR experiment; specifically

within five $T_1$ values.

### 267    2.2 Describing hyperpolarized kinetics with the Bloch-McConnell equations

We now consider the predictions made by using Eq. (19) when simulating the evolution of the $x$, $y$, and

$z$ components of the magnetization of a hyperpolarized sample and the resulting spectrum for a two-site exchange

reaction between solutes A and B. In the previous example the initial condition was $M_z^A(0) = 1.0$ and $M_z^B(0) =$

$0.8$. To extend the Bloch-McConnell formalism to be able to predict the dynamics of a hyperpolarized experiment

we recognize that for the same magnitude of noise in the receiver circuit (although this may not be true for a

hyperpolarized sample) the initial hyperpolarized magnetization is given by:


$$M_{z,hyp} = \eta M_{z,eq}\quad, \tag{21}$$


where $\eta$ is the enhancement factor that varies from one hyperpolarization experiment to another. In the case of
dDNP experiments $\eta \cong 10^4$ is typical, although this depends on the method of hyperpolarization, the solute(s) in
question and a set of physicochemical parameters that are described in detail in e.g., (Ardenkjaer-Larsen et al.,

2015).


### 281    2.2.1 Simulations of hyperpolarized kinetics using Eq. (19)

These were performed with an initial magnetization vector $\mathbf{M}(0) = [0.0, 1.0 \times 10^4, 0, 0, 0]$ while the

equilibrium magnetizations were $M_{z,eq}^A = 1.0$ and $M_{z,eq}^B = 0.8$, as used above. This situation corresponds to an
initial hyperpolarized magnetization $M_{z,hyp}^A(0)$ of only solute A. Chemical shifts were $\Omega^A = 10 \times 2\pi$ rad s$^{-1}$ and
$\Omega^B = -10 \times 2\pi$ rad s$^{-1}$, while relaxation times were increased to represent a hyperpolarized $^{13}$C substrate, $R_{1A} =$
$R_{1B} = 1/60 s^{-1}$ and $R_{2A} = R_{2B} = 1\ s^{-1}$ with the rate constants representing an enzyme mediated cell reaction
$k_1 = k_{-1} = 0.005\ s^{-1}$. Figure 2(a) shows the time evolution of the $z$-components of the magnetization, displaying
the familiar (Day et al., 2007) bi-exponential time dependence of $M_{z,hyp}^A(t)$ and $M_{z,hyp}^B(t)$ magnetizations.

We next simulate the effect of applying the pulse sequence shown in Fig. 2(b) corresponding to a time

course type of experiment with multiple sampling of the magnetization and acquisition of an FID at each time-
point. This is representative of real experiments that have been presented in the literature (Gabellieri et al., 2008;
Hill et al., 2013b). The time delays correspond to a pre-scan delay $\tau$, the duration of the pulse $t_p$ and the duration
of the FID $t_{aq}$. The experiment is repeated $n$ times to sample the entire time course where the temporal resolution
is then given by the total repetition time $TR = \tau + t_p + t_{aq}$ and the total duration of the experiment is given by



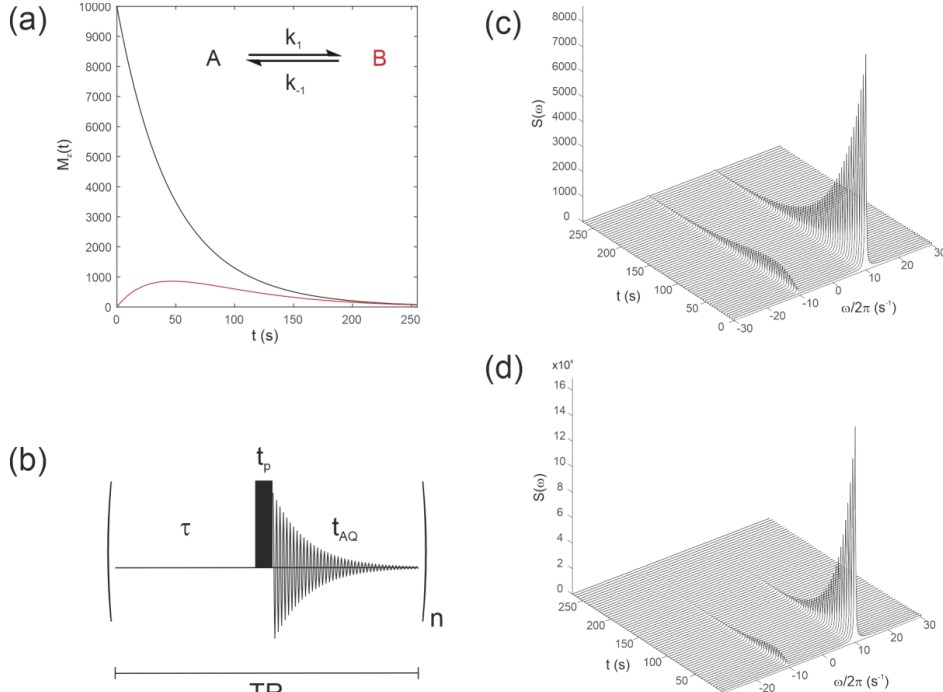

**Figure 2** (a) Simulated evolution of the *z*-components of the magnetization $M_z^A$ and $M_z^B$ for a *hyperpolarized* solute $M_z^A(0) = 1 \times 10^4$ undergoing a two-site exchange reaction, A $\leftrightarrow$ B. Longitudinal relaxation rate constants were $R_{1A} = R_{1B} = 1/60 s^{-1}$ and $R_{2A} = R_{2B} = 1\ s^{-1}$. Rate constants were $k_1 = k_{-1} = 0.005\ s^{-1}$. (b) Simple pulse sequence for acquiring a time course experiment with multiple sampling of the magnetization and acquisition of an FID at each timepoint. (c-d) Waterfall plots of simulated spectra resulting from sequential application of the pulse sequence in (b) for an initial hyperpolarized solute A undergoing two-site exchange with solute B, calculated with a flip angles: (c) $\beta = 1°$; and (d) $\beta = 20°$.

$nTR$. In this experiment we make the assumption that the transverse magnetization from one experiment to the
next is not recovered by the application of a subsequent pulse. This assumption is reasonable provided the
acquisition time is much longer that the time taken for the FID to decay to zero, namely, $t_{aq} \gg T_2^*$.

The influence of this pulse sequence was then calculated, accounting for multiple sampling of the

magnetization. The RF pulse was again specified by $\omega_x = -\gamma B_1 \cos(\pi/2)$ and $\omega_y = -\gamma B_1 \sin(\pi/2)$ with a field
strength of 1.5 kHz, which corresponds to $\omega_y = -\gamma B_1 = -1500 \times 2\pi$ rad s$^{-1}$. Application of an RF pulse tilts
the hyperpolarized magnetization away from the *z* axis by an angle of $\beta$ radians. The magnitude of the observable
transverse magnetization is proportional to $\sin(\beta)$, and the remaining longitudinal magnetization is proportional
to $\cos(\beta)$.

Simulations were performed with the same magnitude of noise as in Fig. 1. The time evolution of the

magnetization was recorded for the pulse sequence shown in Fig. 2(b) with sequential acquisition of 64 spectra,
and a repetition time of $TR = 4.25$ s. The effect of acquiring a time series of spectra with either a flip angle $\beta =$





1°, Fig. 2(c), or $\beta = 20°$, Fig. 2(d), are seen in the stack plots. The pulse length (duration) was $t_p = \beta \, \pi/180\omega_y$.
After a single $\beta = 1°$ pulse applied to $\mathbf{M}(0)$ the magnetization vector was tilted to become $\mathbf{M}(t) =$
$[0.174, 0.000, 9.998, 0.000, 0.000, 0.000] \times 10^3$ prior to acquisition of the FID. This was composed mostly of
$M_z^A$ with a small contribution from $M_x^A$ that arose from excitation by the $\beta = 1°$ pulse; or following a $\beta = 20°$ pulse
the magnetization vector was tilted to become $\mathbf{M}(t) = [3.420, 0.004, 9.397, 0.000, 0.000, 0.000] \times 10^3$, again
comprised mostly of $M_z^A$ but with a greater contribution from $M_x^A$ due to excitation by a pulse with larger value of
$\beta$. Since the magnetization relaxed to its thermal equilibrium state, the hyperpolarized magnetization was
effectively destroyed during application of the RF (sampling) pulse, and it was not re-generated. This may not be
the outcome when non-linear effects such as radiation damping cause recovery of the hyperpolarized signal
(Weber et al., 2019).

The $z$ magnetization after the application of a single RF pulse and delay $TR$ is therefore given by:


$$S(TR) = S(0)\cos(\theta)\exp(-R_1 TR) \quad . \tag{22}$$


Following the application of a series of $n$ RF pulses with a total delay $n\,TR = t$ the signal is given by (Kuchel
and Shishmarev, 2020):

$$S(t) = S(0)\cos^n(\theta)\exp(-R_1 t) \quad . \tag{23}$$


The apparent relaxation time constant of the hyperpolarized signal, including the influence of both the intrinsic
$T_1$ and flip angle correction, is given by (Hill et al., 2013b; Kuchel and Shishmarev, 2020):

$$\exp(-R_{1,app}t) = \cos^n(\theta)\exp(-R_1 t) \quad , \tag{24}$$

$$R_{1,app} = R_1 - \frac{1}{TR}\ln\cos(\theta) \quad . \tag{25}$$


In the previous examples in Figs. 2(c) and 2(d), with a typical $T_1 = 60\,s$ (Keshari and Wilson, 2014)
corresponding to $R_1 = 1.67 \times 10^{-2}\,s^{-1}$ and a $TR = 4.25$ s, the flip angle correction for a $\beta = 1°$ pulse was 3.58
$\times 10^{-5}$, which 'for all intents and purposes', is negligible, giving $R_{1,app} = 1.67 \times 10^{-2}\,s^{-1}$ and $T_{1,app} = 59.87$ s.
Hence, the time dependence of the signal shown in Fig. 2(c) is a robust reflection of the $M_z(t)$ seen in Fig. 2(a).
For $\beta = 20°$ the flip angle correction was $1.46 \times 10^{-2}$ giving $R_{1,app} = 3.13 \times 10^{-2}\,s^{-1}$ and $T_{1,app} = 31.95$ s.
Therefore, for the larger flip angle there was a tradeoff between the increased sensitivity and the corresponding
reduction in $T_{1,app}$ with the more rapid decay of the NMR signal. The time dependence seen in Fig. 2(d) is no
longer a good reflection of the $M_z(t)$ shown in Fig. 2(a). We conclude that when the RF flip angle is small, $< 1°$,
and the magnetization is sampled many times, the flip angle correction is negligible; accordingly, it is ignored in
the next sections.



### 3 Relaxation of hyperpolarized magnetization in $^{13}$C substrates


We now take a detour into relaxation theory to give an overview of the factors that determine the values
of $R_1 = 1/T_1$ of hyperpolarized $^{13}$C solutes in a (bio)chemical system taking into account the main relaxation
mechanisms responsible for the decay of the nuclear magnetization in solution state at temperatures between ~20
to 180°C and static magnetic field strengths between 1 mT to 23.5 T. The spin interactions discussed here are
relevant to the outcome of numerous dissolution-dynamic nuclear polarization (dDNP) experiments.
A master equation for spin systems far from equilibrium based on a Lindblad dissipator formalism has
recently been presented and shown to correctly predict the spin dynamics of hyperpolarized systems (Bengs and
Levitt, 2020). In brief, Eq. (2) is only valid for the high temperature limit and weak order approximation of a spin
system at thermal equilibrium, and therefore the theory accounts for a dependence of relaxation rate constants on
the extent of hyperpolarization. However, we do not pursue this line of enquiry here because for the enzyme
systems studied thus far with dDNP a constant value of $T_1$ has been statistically satisfactory in regression analyses
of the data (Pages et al., 2013; Shishmarev et al., 2018b).
Once a sufficiently high level of nuclear spin polarization has been achieved by implementing dDNP
methodologies (often for $^{13}$C nuclei $P_C > 60\%$) a jet of superheated solvent (*e.g.,* $H_2O$ and/or $D_2O$ at 150-180°C)
is injected directly onto the hyperpolarized sample (Ardenkjaer-Larsen et al., 2003; Wolber et al., 2004). Upon
contact with the warm solvent, the frozen sample rapidly dissolves and is then transferred under the pressure of
helium gas (6-9 bar) to a separate NMR/MRI spectrometer for the detection of hyperpolarized MRS signals, or to
a collection/quality control point for use in biological applications (Comment and Merritt, 2014). There are several
potential issues related to spin relaxation during these processes; and we focus on nuclear spin relaxation in
solution during the sample transfer stage (*i.e.,* subject to changes in magnetic field strength) or situations where a
solute has an altered rotational correlation time (*i.e.,* dependence on temperature or when bound to a protein). This
requires an understanding of the (potentially) large variety of molecular interactions that give rise to nuclear spin
relaxation.
***Dipole-Dipole Couplings (DD).*** The dominant mechanism for the relaxation of nuclear spin
magnetization is often the stochastic modulation of dipole-dipole interactions (couplings) to other nuclei, either
in the same molecule or other molecules, including the solvent, as the molecule re-orientates in solution by
molecular tumbling.
***Chemical Shift Anisotropy (CSA).*** Nuclear spins resonate at different frequencies depending on the
chemical shielding imparted by the local electronic environment and its orientation (a tensor property). The
modulation of the chemical shift tensor by molecular tumbling in solution has a quadratic dependence on the
strength of the static magnetic field and therefore increases markedly with $\mathbf{B}_0$ (Kowalewski and Maler, 2019).
***Paramagnetic Sites.*** Dissolved paramagnetic solutes (often impurities, but they can be purposely added
as required by the experimental design), such as radical agents that remain in the dissolution solvent, molecular
oxygen, and metal ions, which can be deleterious to the nuclear-spin relaxation, particularly in regions of low
magnetic field (Pell et al., 2019; Blumberg, 1960). However, all species can be easily scavenged by co-dissolving
chelating agents in the dissolution medium (Mieville et al., 2010).
***Scalar Relaxation of the Second Kind.*** This mechanism operates when the nuclei of interest have scalar
couplings to neighbouring nuclei that also relax rapidly (Pileio, 2011; Kubica et al., 2014; Elliott et al., 2019). In





dDNP NMR experiments this relaxation mechanism is often enhanced during sample transfer steps through areas
of low magnetic field (Chiavazza et al., 2013; Kubica et al., 2014).
***Spin Rotation.*** The coupling of nuclear magnetization to that of a whole molecule or to mobile parts of
a molecule, *e.g.,* methyl groups, can act as an efficient relaxation mechanism. This mechanism has an unusual
dependence on temperature with the relaxation rate usually increasing at higher temperatures (Matson, 1977).
***Quadrupolar.*** Many molecules of interest in dDNP experiments contain either $^2$H or $^{14}$N nuclei. NMR
relaxation times of such nuclei are often <1 s, and therefore not sufficiently long to be relevant for dDNP
experiments. However, there are two notable exceptions in $^6$Li$^+$ and $^{133}$Cs$^+$ which have small nuclear quadrupole
moments and therefore have intrinsically long $T_1$ values (van Heeswijk et al., 2009; Kuchel et al., 2019).
Derivations of relaxation rate expressions are well established and based on plausible physical models.
For simplicity, we skip the majority of these since they are comprehensively presented by several authors
(Kowalewski and Maler, 2019), and instead we focus on the main results of their analyses. Assuming a two spin
system composed of a $^{13}$C and $^1$H, equations for the $^{13}$C-$^1$H dipole-dipole and the $^{13}$C CSA contributions to the
$^{13}$C longitudinal relaxation rate constant ($R_1$) are given by Keeler (Keeler, 2010):

$$R_{1,DD} = b_{HC}^2 \left[ \frac{3}{20} J(\omega_C) + \frac{1}{20} J(\omega_H - \omega_C) + \frac{3}{10} J(\omega_H + \omega_C) \right] \quad , \tag{26}$$


$$R_{1,CSA} = c^2 \left[ \frac{1}{15} J(\omega_C) \right] \quad , \tag{27}$$


where $b_{HC}$ is the dipole-dipole coupling constant, defined as:

$$b_{HC} = \frac{\mu_0 \gamma_H \gamma_C \hbar}{4\pi r_{HC}^3} \quad , \tag{28}$$


and *c* is the magnitude of the CSA assuming an axially symmetric(al) tensor given by:

$$c = \gamma_C B_0 (\sigma_\parallel - \sigma_\perp) \quad , \tag{29}$$


where $\gamma_H$ and $\gamma_C$ are the magnetogyric ratios of the $^1$H and $^{13}$C spins, respectively, $r_{HC}$ is the internuclear distance
between the $^1$H and $^{13}$C atoms and $\sigma_\parallel$ and $\sigma_\perp$ are the parallel and perpendicular components of the axially
symmetric(al) CSA tensor, respectively.
The so-called spectral density function that is a function of the Larmor frequency, $\omega$, is:

$$J(\omega) = \frac{2\tau_c}{1 + \omega^2 \tau_c^2} \quad , \tag{30}$$


where $\tau_c$ is the rotational correlation time (tumbling motion) of the re-orientating spin-bearing molecule in
solution. The overall longitudinal relaxation rate constant is the sum of these two dominant contributions and is
given by:
$$R_1 = R_{1,DD} + R_{1,CSA} \quad . \tag{31}$$



### 3.1 Relaxation Analysis

It is important (for experimental design purposes) to note the influence that a nearby $^1$H spin has on the $^{13}$C nuclear $T_1$. Figure 3(a) shows the calculated $^{13}$C $T_1$ for a fixed rotational correlation time of $\tau_c = 0.4 \times 10^{-11}$ s (previously reported for glycine in saline at 310 K (Endre et al., 1983)), $^{13}$C CSA $\sigma_\parallel - \sigma_\perp = -98$ppm (previously reported for phosphoenolpyruvate (Bechmann et al., 2004)) and a magnetic field strength of $B_0 = 7$ T as a function of the $^1$H-$^{13}$C internuclear distance $r_{HC}$. Biaxality of the CSA interaction has been ignored here. A rapid rise occurs in $T_1$ as the $^1$H-$^{13}$C internuclear separation increases. In the case of $r_{HC} = 1.09$ Å, which is typical of a $^1$H-$^{13}$C single bond, the $^{13}$C nuclear $T_1$ is predicted to be ~11.4 s. The $^1$H-$^{13}$C dipole-dipole coupling constant scales with $r_{HC}^{-3}$, consequently, the presence of a directly bonded proton significantly shortens the relaxation time constant of the $^{13}$C magnetization. Small molecules containing $^{13}$C atoms that do not have directly bonded $^1$H, or at least $^1$H spins located at significant internuclear distances, are required. Such moieties include the carboxyl group that is present in many low molecular weight metabolites such as pyruvate, lactate, and methylglyoxal (Shishmarev et al., 2018a). At the longer $^1$H-$^{13}$C internuclear distance of 1.45 Å, implying a $^1$H-$^{13}$C dipole-dipole coupling constant of $b_{HC}/2\pi = -10.2$ kHz, a $^{13}$C nuclear $T_1$ of ~60 s is predicted. At very long distances, the $^{13}$C relaxation time constant will tend to that of the CSA relaxation contribution alone.

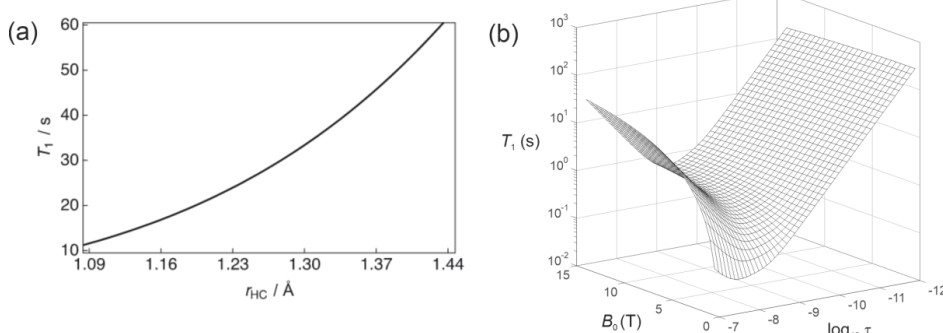

**Figure 3** (a) Simulation of the $^{13}$C nuclear $T_1$ for a two-spin $^1$H-$^{13}$C system as a function of the internuclear distance ($r_{HC}$) with a rotational correlation time $\tau_c = 0.4 \times 10^{-11}$ s, $^{13}$C CSA $\sigma_\parallel - \sigma_\perp = -98$ppm and at a magnetic field strength $B = 7$ T. (b) Dependence of the $^{13}$C nuclear $T_1$ as a function of the magnetic field $B$ and the rotational correlation time $\tau_c$.

The dependence of $R_1$ on temperature and molecular size (*e.g.*, due to binding) scales with the rotational correlation time. Figure 3(b) shows the dependence of the $^{13}$C nuclear $T_1$ ($1/R_1$) as a function of $\tau_c$ and $B_0$ for this 2-spin-1/2 system with $r_{HC} = 1.45$ Å and $\sigma_\parallel - \sigma_\perp = -98$ ppm. In the extreme narrow limit, *i.e.*, $\omega^2\tau_c^2 \ll 1$, the following familiar equations describe the relaxation of $^{13}$C spins under the dipole-dipole and CSA relaxation mechanisms (Kowalewski and Maler, 2019):

$$R_{1,DD} = b_{HC}^2 \tau_c \quad , \tag{32}$$

$$R_{1,CSA} = \frac{2}{15} c^2 \tau_c \quad . \tag{33}$$





In the extreme narrowing regime the $^{13}$C nuclear $T_1$ becomes shorter with increasing magnetic field strength due
to the $B_0^2$ dependence of $R_{1,\text{CSA}}$. At low field strengths, the magnitude of $T_1$ will mostly be attributed to dipole-
dipole relaxation with the nearby $^1$H spin. It is also worth noting that the $^{13}$C $T_1$ follows the usual Lorentzian
spectral density functional dependence on the rotational correlation time. This is clearly seen at high magnetic
field.

**3.2 Molecular Considerations**

The majority of dDNP experiments used to study biological systems employ $H_2O/D_2O$ as the dissolution

solvent. Detection of hyperpolarized NMR/MRI signals typically occurs in a magnetic field range of 1.5-9.4 T,
thus Fig. 3(b) indicates a $^{13}$C nuclear $T_1$ of the order of ~60 s for a carbonyl group, and this is commonly seen in
practice (Shishmarev et al., 2018a). It is important to remember that Eqs. (26-31) provide a greatly simplified
picture of the problem in hand; in reality there are many magnetic nuclei (often within the same molecule) which
contribute to the relaxation of $^{13}$C magnetization. The additional dipole-dipole interactions are likely to be
responsible for differences between predicted and measured $^{13}$C relaxation times, along with the other (more
exotic) signal attenuation mechanisms that are described above.

In a dDNP experiment the dissolution and transfer process can take as long as 15 s; it depends on the

distance to the point of use from the polarizing source; and in clinical applications an additional 30 s can easily
be added for quality control processes. Such requirements place a bound on the usable time in which
hyperpolarized $^{13}$C magnetization must be maintained; and it is typical to expect 45 s to be this limit. Given that
the magnetic field strength "felt" by the hyperpolarized sample can be controlled (to a reasonable extent)
throughout its voyage between the dDNP polarizer and the point of use (Milani et al., 2015), the rotational
correlation time becomes the most important factor that impacts upon the $^{13}$C nuclear $T_1$. Figure 3(b) indicates
that even for a rotational correlation time on the order of $\tau_c = 1 \times 10^{-10}$ s, such as found in proteins in solution
(Wilbur et al., 1976), Eq. (26-31) yields $^{13}$C nuclear $T_1$ relaxation times which are too short to allow practical use
of such samples, *i.e.,* $5 \times T_1 \ll 45$ s, in comparison to the overall time required by a dDNP experiment.

A major parameter that controls the magnitude of the rotational correlation time of a spin-bearing

molecule is its molecular weight ($M_w$). Since $\tau_c \propto M_w$ the rotational correlation time has a noticeable impact on
the $^{13}$C nuclear $T_1$ with even the smallest increase in molecular weight. In order to achieve $^{13}$C nuclear $T_1$ relaxation
times that are sufficiently long to enable hyperpolarized $^{13}$C magnetization to survive the dissolution and transfer
process the $^{13}$C NMR signals must be detectable above the spectral noise for ~45 s. Hence, dDNP samples used
in biological experiments are currently restricted to small molecules (or ions (Kuchel et al., 2019; van Heeswijk
et al., 2009)). For example, the estimate of ~60 s for the $^{13}$C nuclear $T_1$ of the model system described above was
predicted with a rotational correlation time of $\tau_c = 0.4 \times 10^{-11}$ (Endre et al., 1983), and this is sufficiently long for
dDNP experiments.




**3.3 Enzyme Binding**


The worst-case scenario for the model system described in Fig. 3(b) would be a moderate rotational

correlation time of the order of $\tau_c = 1 \times 10^{-8}$ - $1 \times 10^{-10}$ s for which $^{13}$C nuclear $T_1$ relaxation times in the millisecond
regime are predicted. Such correlation times correspond to a system with a molecular weight comparable to that
of an enzyme. If the small molecule (ligand) or ion becomes bound to the enzyme, then it will assume the rotational
correlation time of the higher mass binding partner. In the case of $\tau_c = 1 \times 10^{-9}$ for an enzyme-ligand complex, a
$^{13}$C substrate will have a predicted nuclear $T_1$ of ~276.4 ms at a static magnetic field strength of 7 T. Such a stark
variation in $^{13}$C nuclear $T_1$ values provides good contrast in relaxation-based ligand-protein binding experiments
(Valensin et al., 1982).

**4 Mechanistic description of reaction kinetics of hyperpolarized substrates**


We now consider the interpretation of hyperpolarized dynamics for complex chemical reactions. To help

tease apart the key features of the analysis we begin with some simplifying assumptions. First, in the absence of
an RF pulse Eq. (20) becomes block diagonal, since transverse and longitudinal magnetization are not
interconverted. The evolution of the $z$ magnetization is then dependent only on the initial conditions, $T_1$, and the
rate constants that characterize the chemical exchange. Second, we assume that the $z$ magnetization is sampled
many times with an infinitesimally small flip angle (<<1°) so the longitudinal magnetization decays with its
intrinsic $T_1$ value rather than an apparent $T_{1,app}$ value. Finally, the hyperpolarized magnetization decays to zero,
*i.e.,* the enhancement factor $\eta$ (Eq. (21)) is such that $\mathbf{M}_0$ is greater than $\mathbf{M}_{eq}$ by many orders of magnitude. Thus,
the equilibrium magnetization at $t = \infty$ is effectively zero and it can be ignored in the analysis of real experimental
data.

To reduce clutter in the equations, for all the discussions that now follows, we drop the subscript $z$ since

we hereafter deal only with longitudinal magnetization and denote $M_{z,hyp}^A$ and $M_{z,hyp}^B$ as $A^*(t)$ and $B^*(t)$
corresponding to hyperpolarized magnetization (identified with an asterisk $^*$).

**4.1 Simple first order exchange kinetics of hyperpolarized substrates**


Confining our analysis to the physical subspace that is composed of longitudinal magnetizations, which

describe first-order kinetics of a two-site exchange reaction of hyperpolarized substrates, $A^* \leftrightarrow B^*$, Eq. (20)
simplifies to:

$$\frac{d}{dt}\begin{bmatrix} A^*(t) \\ B^*(t) \end{bmatrix} = \begin{bmatrix} -k_1 - R_1^A & k_{-1} \\ k_1 & -k_{-1} - R_1^B \end{bmatrix}\begin{bmatrix} A^*(t) \\ B^*(t) \end{bmatrix} \quad . \tag{34}$$


Equivalently, Eq. (34) can be expanded to give:

$$\frac{dA^*(t)}{dt} = -k_1 A^*(t) + k_{-1}B^*(t) - R_1^A A^*(t) \quad , \tag{35}$$



$$\frac{dB^*(t)}{dt} = k_1 A^*(t) - k_{-1} B^*(t) - R_1^B B^*(t) \quad, \tag{36}$$


where $k_1$ and $k_{-1}$ denote first-order rate constants, and $R_1^A = 1/T_1^A$ and $R_1^B = 1/T_1^B$ are the longitudinal relaxation
rate constants of A and B, respectively.

Since Eqs. (35) and (36) describe the time evolution of the $z$ magnetizations (that is proportional to

concentration/mass) they do not satisfy the conservation of mass requirement because $d[A^*(t) + B^*(t)]/dt =$
$-R_1^A A^*(t) - R_1^B B^*(t)$ and this tends to zero with time. However, the equations can be recast to specify that the
pools of hyperpolarized substrates relax to form pools of non-polarized substrates A $\leftrightarrow$ B. These pools are denoted
simply by $A(t)$ and $B(t)$ (without the asterisks) as shown in Fig. 4(a). The analogy with radioactive tracers is a
useful one here. A 'hot' pool of radioactive material decays with first order kinetics (half-life) to form a 'cold'
pool of non-radioactive material with the sum of 'hot' and 'cold' being constant.

The kinetics of the non-polarized pools are described by:


$$\frac{dA(t)}{dt} = -k_1 A(t) + k_{-1} B(t) + R_1^A A^*(t) \quad, \tag{37}$$

$$\frac{dB(t)}{dt} = k_1 A(t) - k_{-1} B(t) + R_1^B B^*(t) \quad. \tag{38}$$


Equations (37) and (38) now satisfy conservation of mass, since the rate of change $d[A^*(t) + A(t) + B^*(t) +$
$B(t)]/dt$ is always zero. Note that $A(t)$ and $B(t)$ are not observed in the dDNP NMR experiment; but they are
the counterparts of real concentrations of solute that would be assayable (bio)chemically.

Equations (35-38) can be written as:


$$\frac{d}{dt}\begin{bmatrix} A^*(t) \\ B^*(t) \\ A(t) \\ B(t) \end{bmatrix} = \begin{bmatrix} -k_1 - R_1^A & k_{-1} & 0 & 0 \\ k_1 & -k_{-1} - R_1^B & 0 & 0 \\ R_1^A & 0 & -k_1 & k_{-1} \\ 0 & R_1^B & k_1 & -k_{-1} \end{bmatrix} \begin{bmatrix} A^*(t) \\ B^*(t) \\ A(t) \\ B(t) \end{bmatrix}. \tag{39}$$


We can now appreciate the equivalence between this formalism and conventional chemical reaction kinetics that
are written in terms of molecular concentrations. Furthermore, Eq. (39) can be rewritten as:

$$\frac{d}{dt}\begin{bmatrix} A^*(t) + A(t) \\ B^*(t) + B(t) \end{bmatrix} = \begin{bmatrix} -k_1 & k_{-1} \\ k_1 & -k_{-1} \end{bmatrix} \begin{bmatrix} A^*(t) + A(t) \\ B^*(t) + B(t) \end{bmatrix}, \tag{40}$$


thereby recapturing the conventional form of chemical reaction kinetics for a closed system. Therefore, $A^*(t) +$
$A(t)$ and $B^*(t) + B(t)$ are proportional to $[A(t)]$ and $[B(t)]$, respectively, where the constant of proportionality
depends on the initial experimental conditions, $viz.$, $[A]_0$ and $[B]_0$. In other words, provided $A^*(0) + A(0) = [A]_0$
and $B^*(0) + B(0) = [B]_0$ then the constant of proportionality is 1 and we can equate $A^*(t) + A(t) = [A(t)]$ and
$B^*(t) + B(t) = [B(t)]$. This is a crucial point that we return to below.



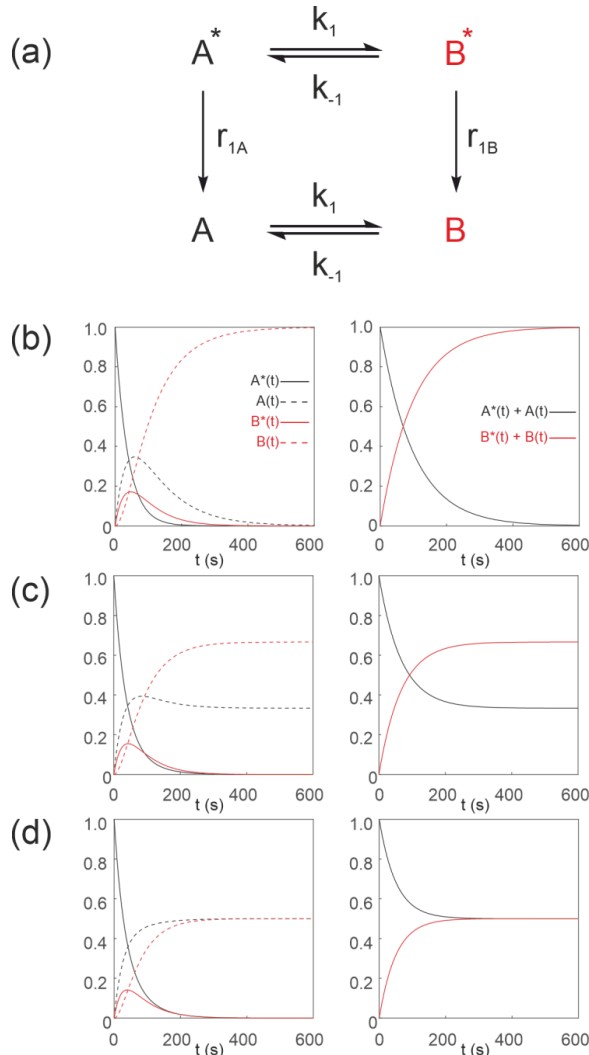

**Figure 4** Simulated first order two-site exchange kinetics of hyperpolarized solutes, A ↔ B, conforming to conservation of mass, assuming initial hyperpolarized magnetization of only solute $A^*(0) = 1$. Longitudinal relaxation rate constants were $R_1^A = R_1^B = 1/60s^{-1}$. The time dependence of $A^*(t), A(t), B^*(t)$ and $B(t)$ (left panel) were calculated numerically using Eq. (35-38) with rate constants (b) $k_1 = 0.01$ s$^{-1}$, $k_{-1} = 0$ s$^{-1}$, corresponding to uni-directional kinetics, (c) $k_1 = 0.01$ s$^{-1}$, $k_{-1} = 0.005$ s$^{-1}$ and (d) $k_1 = 0.01$ s$^{-1}$, $k_{-1} = 0.01$ s$^{-1}$, corresponding to exchange kinetics. The right panel shows plots of the time dependence of $A^*(t) + A(t) = [A(t)]$ and $B^*(t) + B(t) = [B(t)]$.

Figure 4 shows numerical simulations of the time evolution of the system described by Eq. (39) with an
initial magnetization vector $\mathbf{M}(0) = [1, 0, 0, 0]$ that corresponds to only hyperpolarized $A^*(0) = 1$ and
longitudinal relaxation rate constants $R_1^A = R_1^B = 1/60s^{-1}$. The time dependence of $A^*(t), A(t), B^*(t)$ and $B(t)$
were calculated numerically (left panel) for different rate constants: Fig. 4(b), $k_1 = 0.01$ s$^{-1}$, $k_{-1} = 0$ s$^{-1}$,





corresponding to a uni-directional reaction; Fig 4(c), $k_1 = 0.01$ s$^{-1}$, $k_{-1} = 0.005$ s$^{-1}$, corresponding to bi-directional
exchange with an equilibrium constant $K = 2$; and Fig. 4(d), $k_1 = 0.01$ s$^{-1}$, $k_{-1} = 0.01$ s$^{-1}$, also corresponding to bi-
directional exchange with an equilibrium constant $K = 1$. The right column shows plots of the time dependence
of $A^*(t) + A(t)$ and $B^*(t) + B(t)$ that reproduce conventional kinetics of $[A(t)]$ and $[B(t)]$, as required for
mathematical and physical consistency.

The approach used here (as laid out in (Kuchel and Shishmarev, 2020)) enables us to create systems of

differential equations that satisfy conservation of mass and therefore allow a study of the influence of non-
hyperpolarized pools of substrates on reaction kinetics. The approach enables more complicated reaction
mechanisms to be described to allow the inclusion of MR invisible pools of substrates such as $^{12}$C, which are
known to affect the outcome of dDNP experiments *in vivo*. We consider some of these scenarios next.

**4.2 Sequential reaction kinetics of hyperpolarized substrates**
Equation 39 can be extended to compartmental models of arbitrary complexity: Consider a reaction scheme
involving three substrates $A^* \leftrightarrow B^* \leftrightarrow C^*$ which relax through $T_1$ processes to form a pool of non-polarized
substrates $A \leftrightarrow B \leftrightarrow C$, as shown in Fig. 5(a). This is analogous to a system where a solution of hyperpolarized
solute $A^*$ is introduced into the extracellular medium in a cell suspension, is transported into the cells where it is
denoted by $B^*$ and it is subsequently acted upon by an enzyme to form $C^*$. The system of differential equations
that describe the kinetics of this scheme is:

$$\frac{dA^*(t)}{dt} = -k_1 A^*(t) + k_{-1} B^*(t) - R_1^A A^*(t) \quad, \tag{41}$$

$$\frac{dB^*(t)}{dt} = k_1 A^*(t) - k_{-1} B^*(t) - k_2 B^*(t) + k_{-2} C^*(t) - R_1^B B^*(t) \quad, \tag{42}$$

$$\frac{dC^*(t)}{dt} = k_2 B^*(t) - k_{-2} C^*(t) - R_1^C C^*(t) \quad, \tag{43}$$

$$\frac{dA(t)}{dt} = -k_1 A(t) + k_{-1} B(t) + R_1^A A^*(t) \quad, \tag{44}$$

$$\frac{dB(t)}{dt} = k_1 A(t) - k_{-1} B(t) - k_2 B(t) + k_{-2} C(t) + R_1^B B^*(t) \quad, \tag{45}$$

$$\frac{dC(t)}{dt} = k_2 B(t) - k_{-2} C(t) + R_1^C C^*(t) \quad, \tag{46}$$


where we have removed the square brackets that denote molar concentration to avoid some of the clutter.
However, it is important to recall that there is a factor that relates magnetization to concentration, and this is
estimated from the known initial experimental conditions.

Equations (41-46) can be recast in matrix form to give:




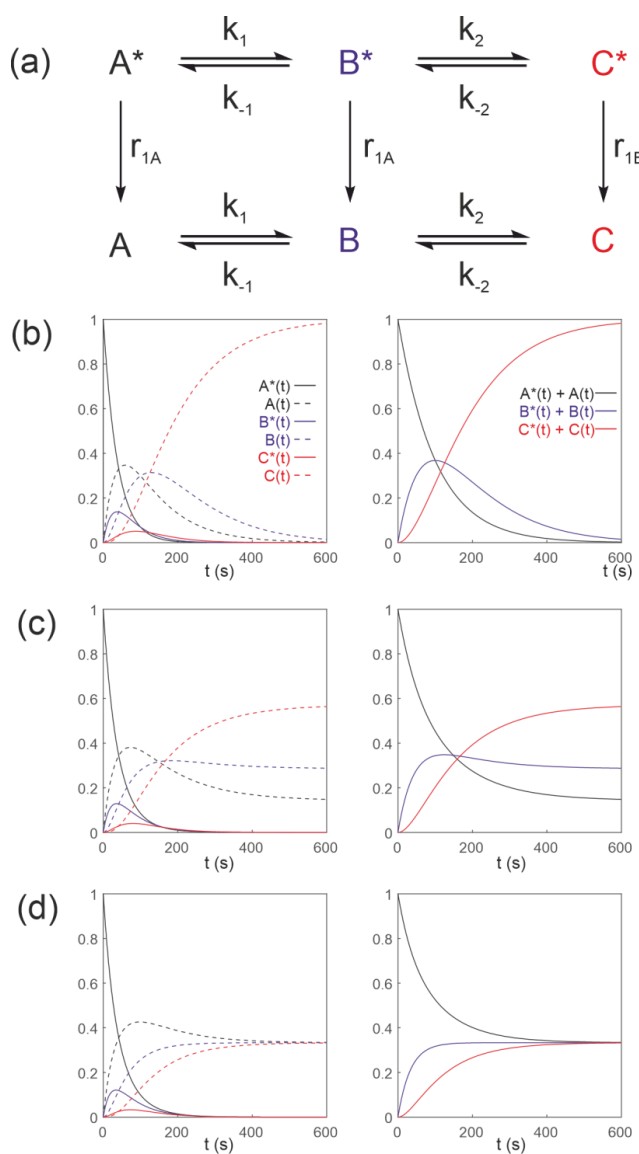

**Figure 5** Simulated first order three-site exchange kinetics of hyperpolarized solutes, A ↔ B ↔ C, conforming to conservation of mass, assuming initial hyperpolarized magnetization of only solute $A^*(0) = 1$. Longitudinal relaxation rate constants were $R_1^A = R_1^B = R_1^C = 1/60 s^{-1}$. The time dependence of $A^*(t)$, $A(t)$, $B^*(t)$, $B(t)$, $C^*(t)$ and $C(t)$ (left panel) were calculated numerically using Eq. (41-46) with rate constants (b) $k_1 = k_2 = 0.01 \ s^{-1}$, $k_{-1} = k_{-2} = 0 \ s^{-1}$, corresponding to uni-directional kinetics, (c) $k_1 = k_2 = 0.01 \ s^{-1}$, $k_{-1} = k_{-2} = 0.005 \ s^{-1}$ and (d) $k_1 = k_2 = k_{-1} = k_{-2} = 0.01 \ s^{-1}$, corresponding to exchange kinetics. The right panel shows plots of the time dependence of $A^*(t) + A(t) = [A(t)]$, $B^*(t) + B(t) = [B(t)]$ and $C^*(t) + C(t) = [C(t)]$.






$$\frac{d}{dt}\begin{bmatrix} A^*(t) \\ B^*(t) \\ C^*(t) \\ A(t) \\ B(t) \\ C(t) \end{bmatrix} = \begin{bmatrix} -k_1 - R_1^A & k_{-1} & 0 & 0 & 0 & 0 \\ k_1 & -k_{-1} - k_2 - R_1^B & k_{-2} & 0 & 0 & 0 \\ 0 & k_2 & -k_{-2} - R_1^C & 0 & 0 & 0 \\ R_1^A & 0 & 0 & -k_1 & k_{-1} & 0 \\ 0 & R_1^B & 0 & k_1 & -k_{-1} - k_2 & k_{-2} \\ 0 & 0 & R_1^C & 0 & k_2 & -k_{-2} \end{bmatrix} \begin{bmatrix} A^*(t) \\ B^*(t) \\ C^*(t) \\ A(t) \\ B(t) \\ C(t) \end{bmatrix} . \tag{47}$$


It is readily verified that Eq. (47) satisfies conservation of mass, since the rate of change $\left( A^*(t) + A(t) + B^*(t) + \right.$
$\left. B(t) + C^*(t) + C(t) \right)/dt = 0$.

Equations (41-46) can be re-written in matrix-vector form as:


$$\frac{d}{dt}\begin{bmatrix} A^*(t) + A(t) \\ B^*(t) + B(t) \\ C^*(t) + C(t) \end{bmatrix} = \begin{bmatrix} -k_1 & k_{-1} & 0 \\ k_1 & -k_{-1} - k_2 & k_{-2} \\ 0 & k_2 & -k_{-2} \end{bmatrix} \begin{bmatrix} A^*(t) + A(t) \\ B^*(t) + B(t) \\ C^*(t) + C(t) \end{bmatrix} , \tag{48}$$


Therefore, provided $A^*(0) + A(0) = [A]_0$, $B^*(0) + B(0) = [B]_0$ and $C^*(0) + C(0) = [C]_0$, then $A^*(t) +$
$A(t) = [A(t)]$, $B^*(t) + B(t) = [B(t)]$ and $C^*(t) + C(t) = [C(t)]$, respectively.

Figure 5 shows the results of numerical integration of Eq. (47) with initial magnetization vector $\mathbf{M}(0) =$

$[1, 0, 0, 0, 0, 0]$ that corresponds to having only hyperpolarized $A^*(0) = 1$ and longitudinal relaxation rate constants
$R_1^A = R_1^B = R_1^C = 1/60 s^{-1}$. The time dependence of $A^*(t)$, $A(t)$, $B^*(t)$, $B(t)$, $C^*(t)$ and $C(t)$ were calculated
(left panel) for different rate constants: Fig. 5(b), $k_1 = k_2 = 0.01 \; s^{-1}$, $k_{-1} = k_{-2} = 0 \; s^{-1}$, corresponding to uni-
directional kinetics; Fig. 5(c), $k_1 = k_2 = 0.01 \; s^{-1}$, $k_{-1} = k_{-2} = 0.005 \; s^{-1}$, corresponding to bi-directional
exchange kinetics; and Fig. 5(d), $k_1 = k_2 = k_{-1} = k_{-2} = 0.01 \; s^{-1}$, also corresponding to bi-directional
exchange kinetics. The right column shows plots of the time dependence of $A^*(t) + A(t)$, $B^*(t) + B(t)$ and
$C^*(t) + C(t)$, which reproduce the conventional chemical kinetics of $[A(t)]$, $[B(t)]$ and $[C(t)]$, as required for
mathematical and physical consistency.

### 4.3 Second-order kinetics of hyperpolarized substrates

We now describe hyperpolarized substrates $A^*(t)$ and $B^*(t)$ reacting with non-hyperpolarized substrates $[C(t)]$
and $[D(t)]$. The system of differential equations that describes these second-order kinetics of $A^* + C \leftrightarrow B^* + D$
with only the hyperpolarized pools relaxing through $T_1$ processes to form a pool of non-polarized substrates $A +$
$C \leftrightarrow B + D$. The reactant concentrations $[C(t)]$ and $[D(t)]$ are common to both pools, as shown in Fig. 6(a). The
relevant system of differential equations (again omitting the square brackets that denote concentration) is:

$$\frac{dA^*(t)}{dt} = -k_1 C(t) A^*(t) + k_{-1} D(t) B^*(t) - R_1^A A^*(t) , \tag{49}$$

$$\frac{dB^*(t)}{dt} = k_1 C(t) A^*(t) - k_{-1} D(t) B^*(t) - R_1^B B^*(t) , \tag{50}$$





$$\frac{dA(t)}{dt} = -k_1 C(t)\, A(t) + k_{-1} D(t) B(t) + R_1^A A^*(t) \quad , \tag{51}$$

$$\frac{dB(t)}{dt} = k_1 C(t)\, A(t) - k_{-1} D(t)\, B(t) + R_1^B B^*(t) \quad , \tag{52}$$

$$\frac{d[C(t)]}{dt} = -k_1 C(t)\left(A^*(t) + A(t)\right) + k_{-1} D(t)\left(B^*(t) + B(t)\right) \quad , \tag{53}$$

$$\frac{d[D](t)}{dt} = k_1 C(t)\left(A^*(t) + A(t)\right) - k_{-1} D(t)\left(B^*(t) + B(t)\right) \quad . \tag{54}$$


As was done above with sets of simultaneous differential equations, Eqs. (49-54) can be cast into matrix-vector
form:

$$\frac{d}{dt}\begin{bmatrix} A^*(t) + A(t) \\ B^*(t) + B(t) \\ C(t) \\ D(t) \end{bmatrix} = \begin{bmatrix} -k_1 C(t) & k_{-1} D(t) & 0 & 0 \\ k_1 C(t) & -k_{-1} D(t) & 0 & 0 \\ -k_1 C(t) & k_{-1} D(t) & 0 & 0 \\ k_1 C(t) & -k_{-1} D(t) & 0 & 0 \end{bmatrix}\begin{bmatrix} A^*(t) + A(t) \\ B^*(t) + B(t) \\ C(t) \\ D(t) \end{bmatrix} \quad . \tag{55}$$


Again, mass is conserved as seen by the fact that $d((A^*(t) + A(t) + B^*(t) + B(t))/dt = 0$ and $d\big(C(t) +$
$D(t)\big)/dt = 0$. Also, recall that provided $A^*(0) + A(0) = [A]_0$, $B^*(0) + B(0) = [B]_0$, $C(0) = [C]_0$ and $D(0) =$
$[D]_0$, then we can make use of the equalities $A^*(t) + A(t) = [A(t)]$, $B^*(t) + B(t) = [B(t)]$, $C(t) = [C(t)]$ and
$D(t) = [D(t)]$, respectively. It is now very evident why we must equate the initial signal with the concentration
via an experimentally estimated scaling factor.

Figure 6 shows numerical simulations of the time evolution of the system of Eqs. (49-54) with initial

magnetization corresponding to the hyperpolarized signal $A^*(0) = 1$ and non-polarized substrates $C(0) = 0.95$
and $D(0) = 0.05$. The longitudinal relaxation rate constants were $R_{1A} = R_{1B} = 1/60 s^{-1}$. The time dependence
of $A^*(t)$, $A(t)$, $B^*(t)$ and $B(t)$ are subject to second order kinetics and were calculated numerically (left panel)
for different rate constants: Fig. 6(b), $k_1 = 0.01\ s^{-1}$, $k_{-1} = 0\ s^{-1}$, corresponding to unidirectional kinetics; Fig.
6(c), $k_1 = 0.01\ s^{-1}$, $k_{-1} = 0.005\ s^{-1}$, corresponding to bi-directional exchange kinetics with an equilibrium
constant $K = 2$; and Fig. 6(d) $k_1 = k_{-1} = 0.01\ s^{-1}$, with an equilibrium constant $K = 1$, also corresponding to bi-
directional exchange kinetics. The right column shows plots of the time dependence of $A^*(t) + A(t)$, $B^*(t) +$
$B(t)$, which capture conventional chemical kinetics of the concentrations of $[A(t)]$ and $[B(t)]$, as required, as
well as the kinetics of the non-polarized reactants $[C(t)]$ and $[D(t)]$.

**4.3.1 An Ersatz solution**
The system of differential equations in Eq. (55), describing a second order reaction can be reduced to one with
pseudo first order kinetics by introducing time-dependent rate constants $k_1'(t) = k_1 C(t)$ and $k_{-1}'(t) = k_{-1}\, D(t)$.
Importantly, the pseudo first order rate constants $k_1'(t)$ and $k_{-1}'(t)$ are now time dependent. This approach has



been used previously (Mariotti et al., 2016) but it constitutes a special case of the more general method described
here, which we advocate.


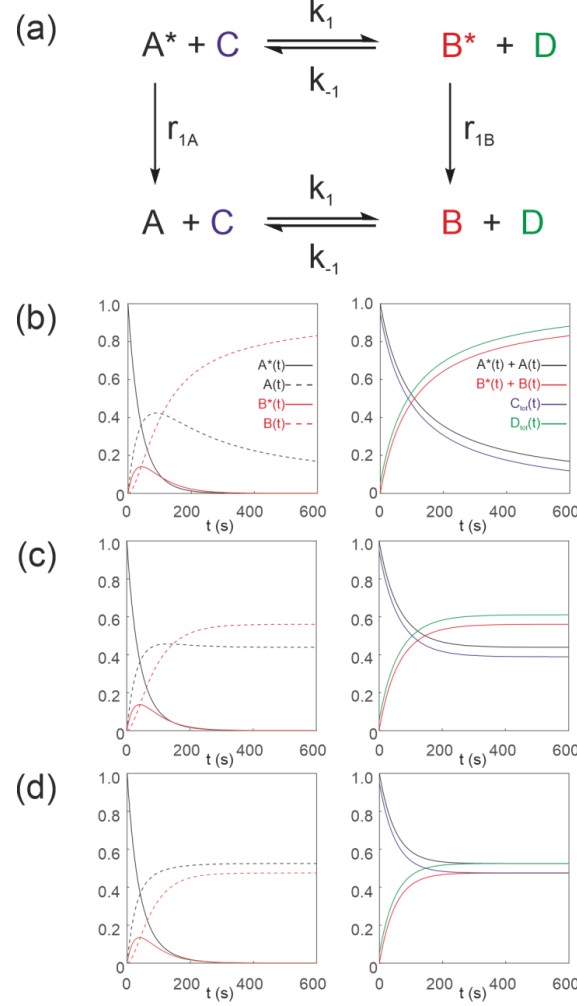

**Figure 6** Simulated second order exchange kinetics of hyperpolarized solutes, $A^* + C \leftrightarrow B^* + D$, conforming to conservation of mass, assuming initial hyperpolarized magnetization of only solute $A^*(0) = 1$. Longitudinal relaxation rate constants were $R_1^A = R_1^B = 1/60 s^{-1}$. The time dependence of $A^*(t)$, $A(t)$, $B^*(t)$ and $B(t)$ were simulated (left panel) using Eqs. (49-54) with rate constants (b) $k_1 = 0.01 \ s^{-1}$, $k_{-1} = 0 \ s^{-1}$, corresponding to uni-directional kinetics (c) $k_1 = 0.01 \ s^{-1}$, $k_{-1} = 0.005 \ s^{-1}$ and (d) $k_1 = k_{-1} = 0.01 \ s^{-1}$, corresponding to exchange kinetics. The right panel shows plots of the time dependence of $A^*(t) + A(t) = [A(t)]$, $B^*(t) + B(t) = [B(t)]$ and non-polarized reactants $[C(t)]$ and $[D(t)]$.






**5 Michaelis-Menten equation for a hyperpolarized substrate**

Next consider an enzyme catalysed reaction with a hyperpolarized substrate. The simplest model involves a hyperpolarized substrate $S^*(t)$ that is in equilibrium with a free enzyme of concentration $[E]_0$ to form a hyperpolarized enzyme substrate complex $ES^*(t)$, which then reacts to form a hyperpolarized product $P^*(t)$. This is followed by release of the free enzyme that is then available for further reactions: $E + S^* \leftrightarrow ES^* \leftrightarrow P^* + E$. All hyperpolarized substrates relax through $T_1$ processes to form non-polarized pools of substrates $E + S \leftrightarrow ES \leftrightarrow P + E$ as shown in Fig. 7(a). The differential equations (again omitting the square brackets denoting concentration) that describe the reaction kinetics are:

$$\frac{dS^*(t)}{dt} = -k_1 E(t)S^*(t) + k_{-1}ES^*(t) - R_1^S S^*(t) \quad , \tag{56}$$

$$\frac{dS(t)}{dt} = -k_1 E(t)S(t) + k_{-1}ES(t) + R_1^S S^*(t) \quad , \tag{57}$$

$$\frac{dES^*(t)}{dt} = k_1 E(t)S^*(t) - k_{-1}ES^*(t) - k_2 ES^*(t) + k_{-2}E(t)P^*(t) - R_1^{ES}ES^*(t) \quad , \tag{58}$$

$$\frac{dES(t)}{dt} = k_1 E(t)S(t) - k_{-1}ES(t) - k_2 ES(t) + k_{-2}E(t)P(t) + R_1^{ES}ES^*(t) \quad , \tag{59}$$

$$\frac{dP^*(t)}{dt} = k_2 ES^*(t) - k_{-2}E(t)P^*(t) - R_1^P P^*(t) \quad , \tag{60}$$

$$\frac{dP(t)}{dt} = k_2 ES(t) - k_{-2}E(t)P(t) + R_1^P P^*(t) \quad , \tag{61}$$

$$\frac{dE(t)}{dt} = -k_1 E(t)\big(S^*(t) + S(t)\big) + (k_{-1} + k_2)\big(ES^*(t) + ES(t)\big) - k_{-2}E(t)\big(P^*(t) + P(t)\big) \quad , \tag{62}$$

where $E(t)$ is the free enzyme, $ES(t)$ is the enzyme-substrate complex, $S(t)$ is the free substrate and $P(t)$ is the free product, with relaxation rate constants $R_1^S$, $R_1^{ES}$ and $R_1^P$, respectively. Note the appearance of the free enzyme $E(t)$ as both a reactant and product; it is regenerated through the reactions that are characterized by the rate constants $k_1$ and $k_{-1}$, and also $k_2$ and $k_{-2}$, thereby being recycled.

Mass is conserved as confirmed by the fact that $d\big(S^*(t) + S(t) + ES^*(t) + ES(t) + P^*(t) + P(t)\big)/dt = 0$ and $d\big(ES^*(t) + ES(t) + E(t)\big)/dt = 0$. Therefore, provided $S^*(0) + S(0) = [S]_0$, $ES^*(0) + ES(0) = [ES]_0$ and $P^*(0) + P(0) = [P]_0$ then $S^*(t) + S(t) = [S(t)]$, $ES^*(t) + ES(t) = [ES(t)]$ and $P^*(t) + P(t) = [P(t)]$, respectively.





### 5.1 Steady state of ES complex


A simplified uni-directional enzyme catalysed reaction is described by setting the reverse rate constant
$k_{-2} = 0$ (see Fig. 7(a)). If it is assumed that a steady-state of $[ES]$ is attained very rapidly then
$d(ES^*(t) + ES(t))/dt = 0$ and we obtain (reverting to using square brackets to denote molar concentration):

$$k_1[E(t)][S^*(t) + S(t)] = (k_{-1} + k_2)[ES^*(t) + ES(t)] \quad . \tag{63}$$


Rearranging Eq. (63) yields the Michaelis constant in terms of hyperpolarized and non-polarized pools of
substrate:

$$K_M = \frac{(k_{-1} + k_2)}{k_1} = \frac{[E(t)][S^*(t) + S(t)]}{[ES^*(t) + ES(t)]} \quad . \tag{64}$$


Calibrating the signals to molar concentrations is important since the signals now relate to a real parameter ($K_M$)
of the enzyme that has units of concentration (typically mM).
Thus, using conservation of enzyme mass, the free enzyme concentration is given by:

$$[E(t)] = [E]_0 - [ES^*(t) + ES(t)] \quad . \tag{65}$$


Then

$$\frac{d([P^*(t) + P(t)])}{dt} = \frac{k_2[E]_0 \, [S^*(t) + S(t)]}{K_M + [S^*(t) + S(t)]} \quad . \tag{66}$$


In other words, this is the standard form of the Michaelis-Menten equation written as a function of both polarized
and unpolarized pools of substrate.

### 5.2 Simulations of Michaelis-Menten reaction

Figure 7(b-c) shows the results of numerical integration of Eqs. (56-62) with an initial hyperpolarized
signal $S^*(0) = 0.001$ (corresponding to a concentration $[S]_0 = 1$ mM via the experimentally determined scaling
factor, which here was set to 1) and enzyme concentration $[E]_0 = 1 \times 10^{-9}$ M. The assigned longitudinal
relaxation rate constants were $R_{1S} = R_{1ES} = R_{1P} = 1/60 s^{-1}$. In the first instance, we set the longitudinal
relaxation times of substrate, enzyme-substrate complex and product to be equal (this is discussed further below).
The reaction rate constants were $k_1 = 1 \times 10^7 \, s^{-1}$, $k_{-1} = 1 \times 10^2 \, s^{-1}$, $k_2 = 5 \times 10^3 \, s^{-1}$, $k_{-2} = 0 \, s^{-1}$, such
that $K_M = 5.1 \times 10^{-4} \, M$ and $V_{max} = 5 \times 10^{-6} \, M \, s^{-1}$. The time dependences of $S^*(t)$, $S(t)$, $P^*(t)$ and $P(t)$ are
shown in Fig. 7(b), left panel, subject to standard uni-directional Michaelis-Menten kinetics; and in Fig. 7(c), left
panel, the time dependence of $ES^*(t)$ and $ES(t)$. The time dependence of $S^*(t) + S(t) = [S(t)]$ and $P^*(t) +$

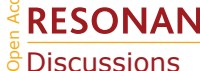

$P(t) = [P(t)]$ are shown in Fig. 7(b), right panel, and $ES^*(t) + ES(t) = [ES(t)]$ and $[E(t)]$ are shown in Fig.
7(c), right panel, which recapture conventional chemical kinetics of $[S(t)]$, $[ES(t)]$, $[P(t)]$ and $[E(t)]$, as required
for mathematical and physical consistency.


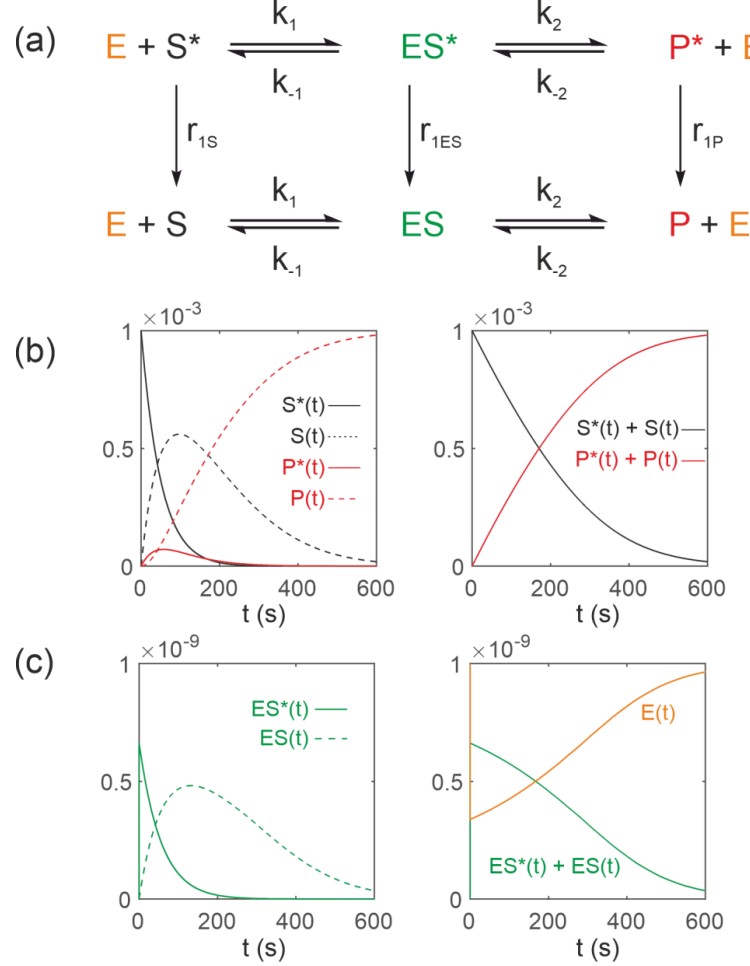

**Figure 7** Simulated Michaelis-Menten kinetics for exchange of hyperpolarized solutes $E + S^* \leftrightarrow ES^* \leftrightarrow P^* + E$ conforming to conservation of mass, assuming initial hyperpolarized magnetization of only solute $S^*(0) = 0.001$ and $[E]_0 = 1 \times 10^{-9}$ M. Longitudinal relaxation rate constants were $R_{1S} = R_{1ES} = R_{1P} = 1/60 s^{-1}$. The reaction rate constants were $k_1 = 1 \times 10^7 s^{-1}$, $k_{-1} = 1 \times 10^2 s^{-1}$, $k_2 = 5 \times 10^3 s^{-1}$ and $k_{-2} = 0 s^{-1}$, such that $K_M = 5.1 \times 10^{-4}$ M and $V_{max} = 5 \times 10^{-6}$ M $s^{-1}$. Left panels: (b) Simulated time dependence of $S^*(t)$, $S(t)$, $P^*(t)$ and $P(t)$; and (c) simulated time dependence of $ES^*(t)$ and $ES(t)$. Right panels: (b) simulated time dependence of $S^*(t) + S(t) = [S(t)]$ and $P^*(t) + P(t) = [P(t)]$; and (c) $ES^*(t) + ES(t) = [ES(t)]$ and $[E(t)]$.





It is worth considering some of the consequences of Eq. (66) when studying enzyme mediated reactions
with hyperpolarized substrates. When the substrate concentration $[S^*(t) + S(t)]$ is much greater than $K_M$ then the
rate of product formation $d([P^*(t) + P(t)])/dt$ is given by $v = k_2[E]_0 = V_{max}$, which is constant (*i.e.*, it is
effectively a zero order reaction with respect to substrate concentration). The enzyme is said to be saturated; its
rate is independent of substrate concentration but $V_{max}$ is proportional to the enzyme concentration $[E]_0$. When
the substrate concentration $[S^*(t) + S(t)]$ is much less than $K_M$ then the rate of product formation
$d([P^*(t) + P(t)])/dt$ is given by $V = k_2[E]_0[S^*(t) + S(t)]/K_M$ and the reaction is effectively first order with
respect to substrate concentration. Nevertheless, the rate is still proportional to $[E]_0$. The kinetics of enzyme
systems, and indeed enzyme kinetics in general, are a composite of the two parameters $K_M$ and $V_{max}$. The influences
on one cannot be distinguished from the other on the basis of time-course experiments alone; separate
measurements that are needed to estimate the total enzyme concentration.
Further simulations were performed to explore the influence of a much shorter value of $T_1^{ES}$ for the
enzyme substrate complex, while $T_1^S$ and $T_1^P$ were unchanged. Even if it were assumed to be very small *viz.*,
$T_1^{ES} = 276.4$ ms the time evolution was indistinguishable from that presented in Fig. 7; the corresponding curves
were superimposable. The signal that resided on the enzyme substrate complex $ES^*$ was 6 orders of magnitude
lower than that of the substrate $S^*$ and product $P^*$. Therefore, the kinetics of signal evolution were dominated by
$T_1^S$ and $T_1^P$ while changes in $T_1^{ES}$ could be ignored. An exception to this analysis might occur if the active site were
next to a paramagnetic centre, such as is found in metalloproteins for which $T_1^{ES}$ could be very much shorter than
predicted (see the relaxation theory section above}.

**5.3 Enzyme inhibition and hyperpolarized substrate kinetics**
Our formalism can be readily extended to account for the influence of a ligand/solute to inhibit an
enzyme. The simplest case is when a solute binds reversibly to the free enzyme E to form an enzyme inhibitor
complex EI; hence, the enzyme becomes unable to bind and react with its substrate S. To describe this scenario,
Eq. 45 is modified to include an additional pathway for the loss of free enzyme:

$$\frac{d[E(t)]}{dt} = -k_1[E(t)][S^*(t) + S(t)] + (k_{-1} + k_2)[ES^*(t) + ES(t)] - k_{-2}[E(t)][P^*(t) + P(t)]$$
$$- k_3[E(t)][I(t)] + k_{-3}[EI(t)] \quad . \tag{67}$$


The model is now extended to include differential equations describing the concentration of the inhibitor $[I(t)]$
and the enzyme-inhibitor complex $[EI(t)]$:

$$\frac{d[I(t)]}{dt} = -k_3[E(t)][I(t)] + k_{-3}[EI(t)] \quad , \tag{68}$$

$$\frac{d[EI(t)]}{dt} = k_3[E(t)][I(t)] - k_{-3}[EI(t)] \quad . \tag{69}$$

Such equations can be incorporated into the Michaelis-Menten equations and we develop this next.





**5.3.1 Types of enzyme inhibition**

There are three commonly encountered types of reversible enzyme inhibition (Kuchel, 2009): (*i*) a *competitive* inhibitor is structurally similar to the substrate and binds preferentially in the active site of the free enzyme, E, thus preventing the substrate from binding and reacting; (*ii*) an *uncompetitive* inhibitor binds only to the enzyme-substrate complex and therefore causes substrate-concentration dependent inhibition; and (*iii*), a *non-competitive* inhibitor binds to both the free enzyme and to the enzyme-substrate complex; it causes a conformational change at the active site that inhibits (or even enhances) the reaction. Such an effect is referred to as allosteric inhibition (or activation).

Accounting for all three scenarios, the free enzyme concentration is given by:

$$[E(t)] = [E]_0 - [EI(t)] - [ES^*(t) + ES(t)] - [ESI^*(t) + ESI(t)] \quad . \qquad (70)$$

Substituting:

$$\alpha = 1 + \frac{[I(t)]}{K_I} \quad \text{and} \quad \alpha' = 1 + \frac{[I(t)]}{K_I'} \quad , \qquad (71)$$

where $K_I = [E(t)][I(t)]/[EI(t)]$ and $K_I' = [ES(t)][I(t)]/[ESI(t)]$, yields:

$$\frac{d([P^*(t) + P(t)])}{dt} = \frac{k_2[E]_0[S^*(t) + S(t)]}{\alpha K_M + \alpha'[S^*(t) + S(t)]} \quad . \qquad (72)$$

The three types of enzyme inhibition can be distinguished by their influence on the kinetic parameters that are estimated in specially designed experiments performed on the enzyme over a range of substrate and inhibitor concentrations (Kuchel, 2009): (*i*) competitive inhibitors cause an increase in apparent $K_M$ value while $V_{max}$ is unchanged; (*ii*) uncompetitive inhibitors cause a reduction in $V_{max}$ while the apparent $K_M$ is unchanged; and (*iii*) non-competitive inhibitors cause both a reduction in $V_{max}$ and an increase in apparent $K_M$.

An additional effect that can be considered is where either the substrate of the reaction $[S(t)]$, or the product of the reaction, $[P(t)]$, acts as the inhibitor, called unsurprisingly 'substrate inhibition' and 'product inhibition', respectively. The relevant enzyme kinetic equations are composed by substituting $[I(t)] = [S^*(t) + S(t)]$ or $[I(t)] = [P^*(t) + P(t)]$ in the above equations (refs).

**6 Cofactors and unlabelled pools – Lactate Dehydrogenase**

We now consider a real system that is of contemporary interest for *in vivo* clinical studies using dDNP. It is lactate dehydrogenase (E.C. 1.1.1.27). Consider the LDH catalysed reaction of a hyperpolarized substrate; it follows an ordered sequential reaction in which E + NADH ↔ E·NADH + Pyr* ↔ E·NAD + Lac* ↔ E + NAD⁺. Again, we assume that relaxation of magnetization occurs through $T_1$ processes to form a pool of reactants E +



NADH $\leftrightarrow$ E·NADH + Pyr $\leftrightarrow$ E·NAD + Lac $\leftrightarrow$ E + NAD$^+$ as shown in Fig. 8(a). The relevant differential
equations used to describe the kinetics are (omitting the square brackets that denote concentration):

$$\frac{dPyr^*(t)}{dt} = -k_2 E.NADH(t)Pyr^*(t) + k_{-2}E.NAD(t)Lac^*(t) - R_1^P Pyr^*(t) \quad , \tag{73}$$

$$\frac{dPyr(t)}{dt} = -k_2 E.NADH(t)Pyr(t) + k_{-2}E.NAD(t)Lac(t) + R_1^P Pyr^*(t) \quad , \tag{74}$$

$$\frac{dLac^*(t)}{dt} = k_2 E.NADH(t)Pyr^*(t) - k_{-2}E.NAD(t)Lac^*(t) - R_1^L Lac^*(t) \quad , \tag{75}$$

$$\frac{dLac(t)}{dt} = k_2 E.NADH(t)Pyr(t) - k_{-2}E.NAD(t)Lac(t) + R_1^L Lac^*(t) \quad , \tag{76}$$

$$\frac{dNADH(t)}{dt} = -k_1 E(t)NADH(t) + k_{-1}E.NADH(t) \quad , \tag{77}$$

$$\frac{dNAD(t)}{dt} = k_3 E.NAD(t) - k_{-3}E(t)NAD(t) \quad , \tag{78}$$

$$\frac{dE.NADH(t)}{dt} = k_1 E(t)NADH(t) - k_{-1}E.NADH(t) - k_2 E.NADH(t)(Pyr^*(t) + Pyr(t))$$
$$+ k_{-2}E.NAD(t)(Lac^*(t) + Lac(t)) \quad , \tag{79}$$

$$\frac{dE.NAD(t)}{dt} = k_2 E.NADH(t)(Pyr^*(t) + Pyr(t)) - k_{-2}E.NAD(t)(Lac^*(t) + Lac(t))$$
$$- k_3 E.NAD(t) + k_{-3}E(t)NAD(t) \quad , \tag{80}$$

$$\frac{dE(t)}{dt} = -k_1 E(t)NADH(t) + k_{-1}E.NADH(t) + k_3 E.NAD(t) - k_{-3}E(t)NAD(t) \quad , \tag{81}$$


where $E(t)$ is the concentration of free enzyme, $NAD(t)$ and $NADH(t)$ are the concentrations of the free co-
factors, $E.NAD(t)$ and $E.NADH(t)$ are the concentrations of the enzyme-cofactor complexes and $Pyr(t)$ and
$Lac(t)$ are the free substrates with relaxation rate constants $R_1^P$ and $R_1^L$, respectively.
Mass is conserved as is confirmed by the fact that $d\big(Pyr^*(t) + Pyr(t) + Lac^*(t) + Lac(t)\big)/dt = 0$.
Enzyme concentration is conserved as is confirmed by $d\big(E.NADH(t) + E.NAD(t) + E(t)\big)/dt = 0$ and
cofactor pools are conserved as is confirmed by $d\big(NADH(t) + NAD(t) + E.NADH(t) + E.NAD(t)\big)/dt = 0$.
Therefore, provided $Pyr^*(0) + Pyr(0) = [Pyr]_0$ and $Lac^*(0) + Lac(0) = [Lac]_0$ then $Pyr^*(t) + Pyr(t) =$
$[Pyr(t)]$ and $Lac^*(t) + Lac(t) = [Lac(t)]$, respectively.
Figure 8(b) shows numerical simulations of the time evolution of the system that is described by Eqs.
(73-81) with initial hyperpolarized signal/concentration (see above for a comment on this aspect) $Pyr^*(t) =$
0.001 and longitudinal relaxation rate constants $R_1^P = R_1^L = 1/60s^{-1}$. The kinetic parameters used for lactate
dehydrogenase were as previously published (Witney et al., 2011; Zewe and Fromm, 1962) for the rabbit muscle
enzyme. Enzyme concentration was $[E]_0 = 1.2 \times 10^{-9}$ M and rate constants $k_1 = 1.03 \times 10^8 \, s^{-1}$, $k_{-1} =$
549 $s^{-1}$, $k_2 = 6.72 \times 10^6 \, s^{-1}$, $k_{-2} = 3.44 \times 10^4 \, s^{-1}$, $k_3 = 842 \, s^{-1}$, and $k_{-3} = 9.12 \times 10^5 \, s^{-1}$. Initial
cofactor concentrations were $[NADH(0)] = 1.0 \times 10^{-4}$ M and $[NAD(0)] = 1.0 \times 10^{-3}$ M to give a $[NAD]/$
$[NADH]$ ratio of 10. In the first instance, endogenous pools of hyperpolarized lactate were set to $Lac^*(0) = 0$ ,
and unpolarized pools of both pyruvate and lactate were made zero, *viz.*, $Pyr(0) = 0$ and $Lac(0) = 0$.


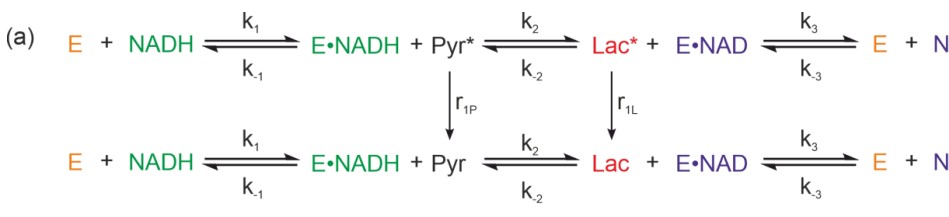

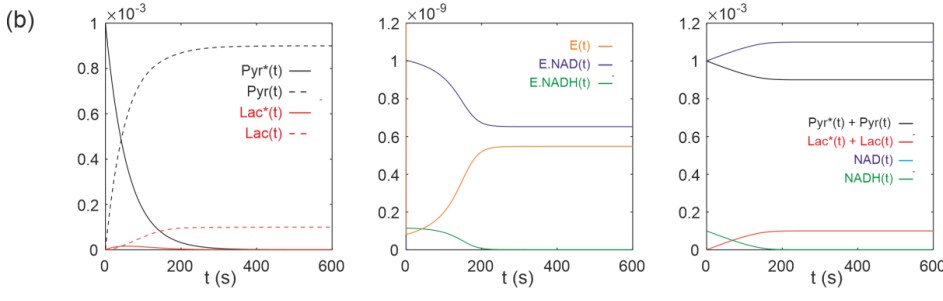

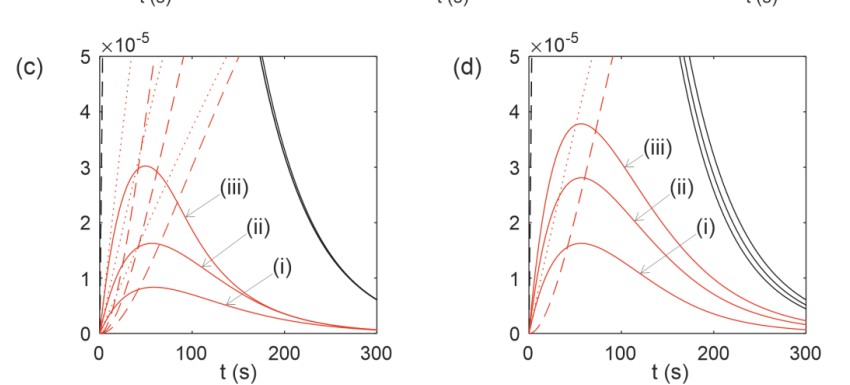

**Figure 8** Simulated kinetics of lactate dehydrogenase for exchange of solutes, $E + NADH \leftrightarrow E \cdot NADH + Pyr^* \leftrightarrow E \cdot NAD$
$+ Lac^* \leftrightarrow E + NAD^+$, conforming to conservation of mass, assuming initial hyperpolarized magnetization of only solute
$Pyr^*(0) = 0.001$ and $[E]_0 = 1.2 \times 10^{-9}$ M. Longitudinal relaxation rate constants were $R_1^P = R_1^L = 1/60 s^{-1}$. Rate
constants were $k_1 = 1.03 \times 10^8 \ s^{-1}$, $k_{-1} = 549 \ s^{-1}$, $k_2 = 6.72 \times 10^6 \ s^{-1}$, $k_{-2} = 3.44 \times 10^4 \ s^{-1}$, $k_3 = 842 \ s^{-1}$ and
$k_{-3} = 9.12 \times 10^5 \ s^{-1}$. Initial cofactor concentrations were $[NADH(0)] = 1.0 \times 10^{-4}$ M and $[NAD(0)] = 1.0 \times$
$10^{-3}$ M. (b) Simulated time dependence $Pyr^*(t)$, $Pyr(t)$, $Lac^*(t)$ and $Lac(t)$ left panel, $[E(t)]$, $[E.NAD(t)]$ and
$[E.NADH(t)]$, middle panel, and $Pyr^*(t) + Pyr(t) = [Pyr(t)]$, $Lac^*(t) + Lac(t) = [Lac(t)]$, $[NAD(t)]$ and
$[NADH(t)]$, right panel. (c) Simulations of the time dependence of $Lac^*(t)$ under the conditions that: $[E]_0 = (i)$
$0.6 \times 10^{-9}$ M; (*ii*) $1.2 \times 10^{-9}$ M; and (*iii*) $2.4 \times 10^{-9}$ M, while all other parameters remained unchanged. (d) Simulations
of the time dependence of $Lac^*(t)$ under the conditions that: $Lac(0) = (i)$ 0 mM; (*ii*) 20 mM; and (*iii*) 40 mM, while
other parameters remained unchanged.



The computed time dependence of polarized and unpolarized pools $Pyr^*(t)$, $Pyr(t)$, $Lac^*(t)$ and
$Lac(t)$ are shown in Fig. 8(b), left panel. The time dependence of $[E(t)]$, $[E.NAD(t)]$ and $[E.NADH(t)]$ are
shown in Fig. 8(b), middle panel. The time dependence of $Pyr^*(t) + Pyr(t) = [Pyr(t)]$, $Lac^*(t) + Lac(t) =$
$[Lac(t)]$, $[NAD(t)]$ and $[NADH(t)]$ are shown in Fig. 8(b), right panel. Several interesting features are evident.
First, the model predicted the expected time dependences of both hyperpolarized pyruvate $Pyr^*(t)$ and its
conversion to $Lac^*(t)$. Under the conditions of the simulation, the free enzyme $[E(t)]$ was rapidly depleted to
form an equilibrium of $[E.NAD(t)]$ and $[E.NADH(t)]$. During the reaction with $Pyr^*(t)$, the equilibrium
position of the enzyme was altered to give a final equilibrium position that could then be appreciated from the
total pools of $Pyr^*(t) + Pyr(t) = [Pyr(t)]$ and $Lac^*(t) + Lac(t) = [Lac(t)]$, which predicts a net conversion
of $[Pyr(t)]$ to $[Lac(t)]$ of ~10%.
Finally, we consider real case scenarios that are reported in the literature. Figure 8(c) shows the situation
where the LDH expression level is altered by the progression of disease (LDH expression is known to be
upregulated in more aggressive cancer phenotypes (Albers et al., 2008)) or down regulated during therapy (Ward
et al., 2010), which can be explored through the value of $[E]_0$. Figure 8(c) shows simulations of the $Lac^*(t)$ signal
under the condition that: $[E]_0 = (i)$ $0.6 \times 10^{-9}$ M; $(ii)$ $1.2 \times 10^{-9}$ M; and $(iii)$ $2.4 \times 10^{-9}$ M, while all other
parameters remained unchanged, relative to those used for Fig. 8(b). It is apparent that increased enzyme
expression leads to an increase in the apparent rate of conversion of $Pyr^*(t)$ to $Lac^*(t)$ even in the absence of a
change in enzyme activity, as seen in real experiments. Another situation that is frequently encountered are
changes in the pool size of endogenous lactate, for example in response to hypoxia, which can be explored through
the parameter $Lac(0)$. Figure 8(d) shows simulations of the $Lac^*(t)$ signal under the conditions that: $Lac(0) =$
$(i)$ 0 mM; $(ii)$ 20 mM; and $(iii)$ 40 mM, while all other parameters remained unchanged, relative to those used to
generate Fig. 8(b). The model therefore predicts that an increased pool of endogenous unpolarized lactate leads to
an increase in the apparent rate of conversion of $Pyr^*(t)$ to $Lac^*(t)$, as reported widely in the literature (Day et
al., 2007).

**7 Conclusion**
We have described an approach to formulating the kinetic master equations that describe the time evolution of
hyperpolarized $^{13}$C NMR signals in reacting (bio)chemical systems, including enzymes with two or more
substrates, and various enzyme reaction mechanisms as classified by Cleland. The modelling can be the basis of
simulating many pertinent features that are seen in dDNP experiments. Derivation of the Michaelis-Menten
equation in the context of dDNP experiments illustrates why formation of a hyperpolarized enzyme-substrate
complex does *not* cause an appreciable loss of the signal from the substrate or product. It was also able to answer
why the concentration of an unlabelled pool of substrate, for example $^{12}$C lactate, causes an increase in the rate of
exchange of the $^{13}$C labelled pool, and to what extent the equilibrium position of an enzyme-catalyzed reaction,
for example LDH, is altered upon adding hyperpolarized substrate. The formalism described here should
contribute to a fuller mechanistic understanding of the time courses derived from dDNP experiments and will be
relevant to ongoing clinical applications using dDNP.



**Author contributions**

All authors planned the research, conducted the research and wrote the paper.

**Competing interests**

The authors declare that they have no conflict of interest.

**Acknowledgements**

The work was supported by the NIHR Biomedical Research Centre at Guy's and St Thomas' NHS Foundation
Trust and KCL; the Centre of Excellence in Medical Engineering funded by the Wellcome Trust and EPSRC (WT
203148/Z/16/Z) and the BHF Centre of Research Excellence (RE/18/2/34213). PWK's work was supported by
and Australian Research Council Discovery Project Grant, DP190100510. SJE was supported by ENS-Lyon, the
French CNRS, Lyon 1 University and the European Research Council under the European Union's Horizon 2020
research and innovation program (ERC Grant Agreements No. 714519 / HP4all).



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
