# Peer review of "Extended Bloch-McConnell equations for mechanistic"

_Magnetic Resonance, 2021_

## Referee Comment (RC1)

This paper describes dissolution DNP-NMR experiments for a variety of reaction schemes using the Bloch-McConnell equations. The topic of the paper is very timely, given the current intense interest in DNP approaches; however, in present form, the main contributions of the paper are difficult to discern.

1. A great deal of the paper recapitulates the work published as:

Kuchel, P. W., and Shishmarev, D.: Dissolution dynamic nuclear polarization NMR studies of enzyme kinetics: Setting up differential equations for fitting to spectral time courses, J. Magn. Reson. Open, 1, 100001, doi.org/10.1016/j.jmro.2020.100001, 2020.

and at least to this reviewer, the novel aspects of the present paper are not clearly distinguished from the earlier work. This could be addressed by an appropriate summary in the conclusion.

I do not understand the statements at line 260-265:

> The equilibrium constant was fixed so that $K = k_1/k_{-1} = 2$; hence the system was not at chemicalcequilibrium at $t = 0$ s. The simulations highlight an important point: In the absence of exchange the Bloch-McConnell equations predict the recovery of the z magnetizations back to their equilibrium values $M^A_{z,eq}$ and $M^B_{z,eq}$ while under conditions of fast exchange this no longer holds, and a non-equilibrium system will rapidly return to its chemical equilibrium, not to its thermal equilibrium, within the timescale of the NMR experiment; specifically within five $T_1$ values.

If the system is not at chemical equilibrium, $M^A_{z,eq}$ and $M^B_{z,eq}$ are not constants as represented in Eq. 19. Rather, these quantities become time-dependent, for example:

$$M^A_{z,eq}(t) = M_0 \frac{[A(t)]}{[A(t)]+[B(t)]}$$

$$M^B_{z,eq}(t) = M_0 \frac{[B(t)]}{[A(t)]+[B(t)]}$$

as shown in Eq. 2.4.18 of Ernst et al. (Ernst, R. R., Bodenhausen, G., and Wokaun, A.: Principles of Nuclear Magnetic Resonance in One and Two Dimensions, Clarendon Press, Oxford, 1987). Thus, it is unclear what is shown in Fig. 1 (which the text indicates is based on Eq. 19) and hence what is meant by the above statement.

Section 4 of the paper expands the dimensionality of the Bloch-McConnell equations to include the molecules that are not hyperpolarized by the DNP. Thus A*(t) represents hyperpolarized species and A(t) represents the unpolarized species, with A*(t) + A(t) = [A(t)] is the total concentration of A at time t (and similarly for B). The authors then numerically solve the four coupled differential equations. This is somewhat misleading in my view. Applying a similarity transformation to Eq. 39 with the matrix:

$$S = \begin{bmatrix} 1 & 0 & 0 & 0 \\ 0 & 1 & 0 & 0 \\ 1 & 0 & 1 & 0 \\ 0 & 1 & 0 & 1 \end{bmatrix}$$

separates the four equations to two independent sets of equations. One set is a pair of coupled differential equations for A*(t) (Eq. 37) and B*(t) (Eq. 38), describing the evolution of the hyperpolarized magnetization and another set (Eq. 40) describing the evolution of [A(t)] and [B(t)]. That is, the hyperpolarized magnetization just evolves according to the normal Bloch-McConnell equations and the chemical system evolves to chemical equilibrium, as expected (note that this situation would be more complicated if time-dependent steady state magnetizations were included as above, but these terms are small and neglected in this section of the paper, given the large DNP enhancements). A four-dimensional set of equations do not need to be solved. I have not checked whether any of the other more complicated schemes presented in the rest of the paper can be similarly decomposed, but it is essential that the authors clarify this issue.

Minor:

Eqs. 19 and 20 seem to be missing a leading minus sign on rhs.

---

## Author Response (AR1)

Referee 1

This paper describes dissolution DNP-NMR experiments for a variety of reaction schemes using the Bloch-McConnell equations. The topic of the paper is very timely, given the current intense interest in DNP approaches; however, in present form, the main contributions of the paper are difficult to discern.

We thank the reviewer for their supportive words and finding the topic to be of current interest. We also thank the reviewer for their valuable comments, addressed below, and hope that our additional analysis helps to clarify the paper.

1. A great deal of the paper recapitulates the work published as:

Kuchel, P. W., and Shishmarev, D.: Dissolution dynamic nuclear polarization NMR studies of enzyme kinetics: Setting up differential equations for fitting to spectral time courses, J. Magn. Reson. Open, 1, 100001, doi.org/10.1016/j.jmro.2020.100001, 2020.

and at least to this reviewer, the novel aspects of the present paper are not clearly distinguished from the earlier work. This could be addressed by an appropriate summary in the conclusion.

We appreciate that our current paper leads on from that previously published. This was commented on in our Abstract. We have added the reference in the abstract to clarify this point. However, our analysis includes a number of aspects not previously considered. Notably, we extend the analysis to include the influence of enzyme cofactors. We use the framework to simulate the kinetics of lactate dehydrogenase, not previously reported using this analysis, and consider the influence of enzyme concentration and the influence of unlabelled lactate on the kinetics of the hyperpolarized pool. Thus, our work is of direct relevance to ongoing clinical trials using this method.

I do not understand the statements at line 260-265:

The equilibrium constant was fixed so that $K = k_1/k_{-1} = 2$; hence the system was not at chemical equilibrium at t = 0 s. The simulations highlight an important point: In the absence of exchange the Bloch-McConnell equations predict the recovery of the z magnetizations back to their equilibrium values $M_{z,eq}^A$ and $M_{z,eq}^B$ while under conditions of fast exchange this no longer holds, and a nonequilibrium system will rapidly return to its chemical equilibrium, not to its thermal equilibrium, within the timescale of the NMR experiment; specifically within five T1 values.

If the system is not at chemical equilibrium, $M_{z,eq}^A$ and $M_{z,eq}^B$ are not constants as represented in Eq. 19. Rather, these quantities become time-dependent, for example:

$$M_{z,eq}^A(t) = M_0 \frac{[A(t)]}{[A(t)] + [B(t)]}$$

$$M_{z,eq}^B(t) = M_0 \frac{[B(t)]}{[A(t)] + [B(t)]}$$

as shown in Eq. 2.4.18 of Ernst et al. (Ernst, R. R., Bodenhausen, G., and Wokaun, A.: Principles of Nuclear Magnetic Resonance in One and Two Dimensions, Clarendon Press, Oxford, 1987). Thus, it is unclear what is shown in Fig. 1 (which the text indicates is based on Eq. 19) and hence what is meant by the above statement.

This is exactly the point that we were making in Fig. 1. Under conditions of exchange the longitudinal magnetization returns to its chemical equilibrium as shown in Eq. 2.4.18 of Ernst et al. not to its initial magnetic equilibrium given by $M_{z,eq}^A$ and $M_{z,eq}^B$. We have modified this statement as follows and hope this adds clarity to the simulations performed:

"The simulations highlight an important point: In the absence of exchange the Bloch-McConnell equations predict the recovery of the z magnetizations back to their magnetic equilibrium values $M_{z,eq}^A$ and $M_{z,eq}^B$ while under conditions of fast exchange this no longer holds, and a nonequilibrium system will rapidly return to its chemical equilibrium, not to its initial thermal equilibrium $M_{z,eq}^A$ and $M_{z,eq}^B$, …"

Section 4 of the paper expands the dimensionality of the Bloch-McConnell equations to include the molecules that are not hyperpolarized by the DNP. Thus A*(t) represents hyperpolarized species and A(t) represents the unpolarized species, with A*(t) + A(t) = [A(t)] is the total concentration of A at time t (and similarly for B). The authors then numerically solve the four coupled differential equations. This is somewhat misleading in my view. Applying a similarity transformation to Eq. 39 with the matrix:

$$S = \begin{bmatrix} 1 & 0 & 0 & 0 \\ 0 & 1 & 0 & 0 \\ 1 & 0 & 1 & 0 \\ 0 & 1 & 0 & 1 \end{bmatrix}$$

separates the four equations to two independent sets of equations. One set is a pair of coupled differential equations for A*(t) (Eq. 37) and B*(t) (Eq. 38), describing the evolution of the hyperpolarized magnetization and another set (Eq. 40) describing the evolution of [A(t)] and [B(t)]. That is, the hyperpolarized magnetization just evolves according to the normal Bloch-McConnell equations and the chemical system evolves to chemical equilibrium, as expected (note that this situation would be more complicated if time-dependent steady state magnetizations were included as above, but these terms are small and neglected in this section of the paper, given the large DNP enhancements). A four-dimensional set of equations do not need to be solved. I have not checked whether any of the other more complicated schemes presented in the rest of the paper can be similarly decomposed, but it is essential that the authors clarify this issue.

We agree with the Referee that this separation of equations using alternative basis vector is possible for all the situations considered in our manuscript. We have consequently performed this transformation throughout in the revised manuscript to clarify this point.

However, the separation of the equations into independent sets of equations only occurs with first order kinetics. In the first order case, we agree that a four-dimensional set of equations does not need to be solved; for example the situation of an exchange reaction A* $\longleftrightarrow$ B* reduces to the standard Bloch McConnell equations and the evolution described by the two-dimension DEs is identical to the four-dimensional. Similarly, for the first order situation A $\longleftrightarrow$ B $\longleftrightarrow$ C.

For second order kinetics the transformation can still be performed but the equations are no longer independent. Considering the example of a hyperpolarized reaction A* + C $\longleftrightarrow$ B* + D and a corresponding non-polarized pool A + C $\longleftrightarrow$ B + D. The pseudo first order rate constants for A and B are dependent on C and D, and vice versa the pseudo first order rate constants for C and D are dependent on (A* + A) and (B* + B). Thus the 'cold' pool of A and B will influence the kinetics of A* and B* via the involvement of C and D and cannot be separated in the same manner as for first order kinetics. The full system of differential equations must be calculated simultaneously to simulate the kinetics.

We have added a comment highlighting the problem on p25:

"However, we now encounter a problem. The pseudo rate constants for the reactions of [C(t)] and [D(t)] are now given by $k'_1(t) = k_1\big(A^*(t) + A(t)\big)$ and $k'_{-1}(t) = k_{-1}\big(B^*(t) + B(t)\big)$, respectively. The time-dependent pseudo first order rate constants are dependent on the concentrations of both 'hot' *and* 'cold' pools. In turn the pseudo first order rate constants for $A^*(t)$ and $B^*(t)$ are $k'_1(t) = k_1 C(t)$ and $k'_{-1}(t) = k_{-1} D(t)$. Thus, the kinetics of the 'hot' pools $A^*(t)$ and $B^*(t)$ become dependent on the kinetics of the 'cold' pools $A(t)$ and $B(t)$. This is of particular relevance (as highlighted in Kuchel and Shishmarev, 2019) when extending the equations to describe enzyme kinetics…"

Minor:

Eqs. 19 and 20 seem to be missing a leading minus sign on rhs.

We thank the Referee for spotting this typo, which we have corrected.

Referee 2

In terms of scientific impact, discussions (and re-discussions) of the kinetics of hyperpolarised magnetisation in presence of chemical exchange and relaxation should be of interest to the community. The paper attempts to clarify two questions that are glossed over in many dissolution-DNP papers: i) substrate-enzyme binding effect on the relaxation rate constants of the substrate and ii) alteration of the kinetics of isotope-labelled product polarisation by an unlabelled pool substrate or product. Even if the paper were not entirely original, the thorough discussion of these topics makes it worth reading.

We thank the reviewer for their positive comments on our work. Indeed it was our aim to give a more complete description of the factors leading to the observed kinetics in hyperpolarized experiments from a mechanistic view point.

Could the equations be presented as Polarisation(time)*Substrate_Concentration(time) ? This way, two variables that obey to different kinetic rate constants could be separated.

Unfortunately, this is not possible in a reacting system. The eigenvalues are linear combinations of both T1 relaxation and the forwards and reverse rate constants k. Therefore, polarization is transferred from one molecular species to the other. In our approach the sum of hyperpolarized and its non-polarized counterpart are explicitly declared to be equal to concentration $A^*(t) + A(t) = [A(t)]$ and therefore polarization does not enter into the models that we present.

In terms of presentation, the paper would gain by exploring some cases such as $R_1 \gg k_{1,-1}$ $R_1 \ll k_{1,-1}$, passage through a membrane before enzymatic conversion, or high enzyme concentrations. In the latter case, equations seem to predict an abrupt decrease of product magnetisation levels, a discussion would be welcome.

We thank the reviewer for this suggestion. Indeed, there is a rather large parameter space that could be further explored through the simulations. At risk of over burdening the current work we defer to the referee's final comment to include the Matlab scripts in a Supplemental information. We have now included this in the revision and therefore prefer to leave this exploration of different parameters to the readers.

However, we note the referee's observation that product magnetization levels decrease at high enzyme concentration. This is a consequence of our simulation parameters. In this case the concentration of [NADH(0)] is limiting and becomes exhausted during the time course (at about 200s in Figure 8). Thus, the enzyme activity switches off due to depletion of cofactor. We have made a comment to this effect (lines 808-811)

References are not complete, at least papers where experimental data was acquired on the kinetics of substrates used as examples in this article should be cited.

We have added references to data published on hyperpolarized [1-$^{13}$C] pyruvate kinetics in cellular systems (lines 812-814).

Minor aspects:

lines 282-287: if k_-1 = k_-1, is the equilibrium value M_z,eq,A/M_z,eq,B = 1/0.8 ?

The starting equilibrium value in the simulation is $M_z^A(0) = 1.0$ and $M_z^B(0) = 0.8$. However, since $k_{-1} = k_1$ the final equilibrium magnetization is $M_z^A(0) = 0.9$ and $M_z^B(0) = 0.9$. The chemical equilibrium is different to the initial thermal equilibrium.

Figure 8: Equations in (a) appear truncated.

We thank the reviewer for spotting this. It has been corrected.

Necessary improvements:

Supporting material containing the Matlab program with appropriate comments should be made available to the readers.

We thank the reviewer for this suggestion and have included a Supplemental Info in the revision that includes all Matlab scripts used in the simulations.